# The π-trap approach for obtaining crystal structure data of inherently amorphous cluster compounds

Yaofeng Wang [1], Niklas Rinn[1], Kevin Eberheim[2], Ferdinand Ziese [2], Jan Christmann [1], Arijit Jana[1], Simon Nier[1], Nils W. Rosemann [1,3], Simone Sanna [2] & Stefanie Dehnen [1] ✉

Single crystal diffraction is one of the most common and powerful tools for structural elucidation. However, obtaining single crystals of adequate size and quality is not always trivial. The "crystalline sponge" method has been used for crystallizing intrinsically amorphous compounds inside a metal organic framework,[1–4] but its application is limited by the size and stability of the pores within the networks. Here, we report the use of π–π interactions between $C_{60}$ and nanometer-sized molecules that by themselves do not form crystalline compounds. Using this "π-trap" approach, we successfully crystallized adamantane-like clusters exhibiting extreme nonlinear optical properties, which so far resisted any attempt for crystallization. $C_{60}$···cluster interactions enabled long-range order, so the clusters' molecular structures could be precisely determined. Spectroscopy and quantum chemical studies showed that clusters and $C_{60}$ behave like being dissolved in each other. This method should be applicable to all kinds of amorphous compounds that undergo π–π interactions.

Single-crystal X-ray diffraction (SCXRD) analysis provides the most precise method for determining the structures of natural and synthetic molecules, making it an indispensable tool for chemists. Naturally, it requires the samples to be obtained in single crystalline form. However, many compounds are intrinsically amorphous, and therefore their structures cannot be subjected to SCXRD analysis. In 2010, Fujita and coworkers introduced a method in which porous coordination networks are used as "crystalline sponges" that absorb target molecules from solution and orient them in a uniform fashion within the crystalline network, which enabled the determination of their molecular structures at atomic resolution by SCXRD[1–4]. While this method does not require the crystallization of the target compound by itself, it requires that the networks' pores have a suitable size and stability to accommodate the guest molecules. Therefore, this technique has been mostly applied to smaller molecules, like 2,6-diisopropylaniline,

guaiazulene, santonin, or chain-like molecules, such as miyakosyne, which comfortably fit into the pores. Larger guests, like fullerene molecules, could be included too, yet in this case, no crystal data were obtained[3]. To the best of our knowledge, the method has so far been used mostly for (bio-)organic molecules or organometallic complexes. In contrast, methods to precisely determine the molecular structures of amorphous compounds that are comprised of (polyhedral) inorganic cluster molecules, have remained elusive to date.

Recently, adamantane-type organic-inorganic hybrid clusters with a general composition of $[(RSn)_4E_6]$ (R = organic substituent, E = S, Se), such as $[(PhSn)_4S_6]$ (**A**, Fig. 1) or $[(PhSn)_4Se_6]$ (**B**), have attracted significant attention due to their (extreme) nonlinear optical (NLO) properties[5,6]. Particularly noteworthy is the phenomenon of continuous-wave infrared laser-induced directional white-light generation (WLG). This has been demonstrated to work exclusively for

[1]Institute of Nanotechnology, Karlsruhe Institute of Technology, Karlsruhe, Germany. [2]Institut für Theoretische Physik and Center for Materials Research (LaMa), Justus-Liebig-Universität Gießen, Gießen, Germany. [3]Light Technology Institute, Karlsruhe Institute of Technology, Karlsruhe, Germany. ✉e-mail: stefanie.dehnen@kit.edu

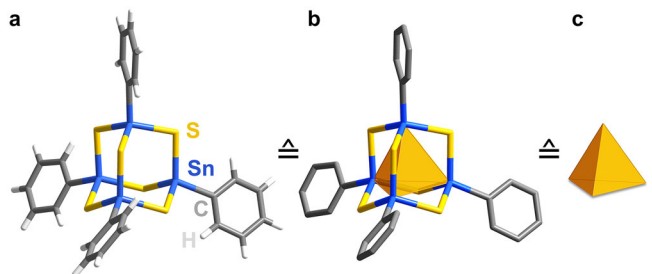

**Fig. 1 | Structural model of the adamantane-type cluster in the amorphous compound [(PhSn)₄S₆] (A) and its simplified representations. a** Full cluster structure (from calculations with DFT-PBE). **b** Simplified representation with H atoms omitted and the (nonbonded) {Sn₄} motif highlighted by a tetrahedron. **c** Even more simplified representation by the inner tetrahedral {Sn₄} motif only.

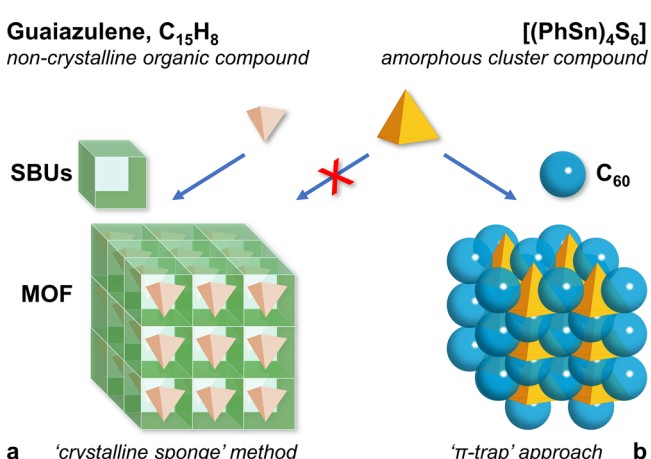

**Fig. 2 | Schematic illustration of the "crystalline sponge"[1,2] method versus the "π-trap" approach.** The approaches are shown on the examples of the organic compound guaiazulene, $C_{15}H_{18}$[1,2], represented by a light brown tetrahedron, and [(PhSn)₄S₆] (**A**), represented by its inner tetrahedral {Sn₄} motif (see Fig. 1c). **a** The "crystal sponge" method applied to smaller-size organic compounds included in the cages formed by the secondary building units (SBUs) of a metal-organic framework (MOF). **b** The "π-trap" approach leading to crystalline compounds accommodating organotetrel chalcogenide cluster molecules between $C_{60}$ spheres.

rigorously amorphous [(RSn)₄E₆] compounds containing electron-rich organic substituents like phenyl (Ph), styryl (Sty), or cyclopentadienyl (Cp), although the physical mechanism behind this phenomenon is still largely ununderstood[7,8]. The structural conformations of the corresponding [(RSn)₄E₆] cluster molecules have been suggested by theoretical studies, also in combination with X-ray and neutron scattering experiments, but could never be verified or supported crystallographically—as the amorphous habitus is intrinsic to those materials[9–11]. Notably, related clusters, such as [(MeSn)₄S₆][5] or [(PhSi)₄S₆] (**C**)[5], form crystalline solids and show strong second-harmonic generation (SHG) instead of WLG.

To date, it is still unclear why the amorphous habitus is required for the WLG phenomenon, and it has also not been clarified until today which parameters control that some of these highly related cluster compounds crystallize while other ones remain rigorously amorphous. There have been suggestions for answering the latter question based on theoretical studies of cluster pair models, which suggested more pronounced (directed) substituent···substituent interactions for compounds featuring smaller cluster cores (e.g., {Si₄S₆}), whereas for larger ones (e.g., {Sn₄S₆}), one observes a dominance of (rather isotropic) interactions of the polyhedral cluster cores[12]. Combined scattering and reverse Monte-Carlo studies revealed information about cluster assemblies, and also suggested significant statistical distortions of the cluster cores, with an increased tendency for larger (and softer) cores[13]. However, a crystallographic proof of all of those hypotheses has still been elusive. Experimental and theoretical physicists have been trying to find answers to the former question since the first observation of the WLG phenomenon, for which a full picture of the amorphous compounds' characteristics—including the essentially inaccessible structural data—is critical[14].

We therefore aimed at identifying a tool for obtaining this data to get knowledge of, and possibly confirm, the reasons for an amorphous versus crystalline habitus. This would lay the foundation for eventually finding an explanation for the physical phenomenon in the near future and altogether ultimately allow to design and control the compounds' habitus and habitus-dependent properties. A rather obvious idea was to apply Fujita's "crystalline sponge method"[1,2]. However, as indicated in the group's seminal publications, the applicability of the technique is limited by the maximum cross-section of pores in metal-organic frameworks (MOFs), which is typically below 1 nm in the materials applied to this technique. MOFs with larger pores exist[15], but they tend to collapse during the process. Unsuccessful attempts with polyhedral [(RSn)₄E₆] molecules confirmed that their outer diameters of ~1–1.5 nm prevent this approach. So, for obtaining precise structural data of cluster molecules that form intrinsically amorphous powders, another methodology is needed.

Here, we report about the introduction of the "π-trap" as a method, which makes use of the cocrystallization of cluster molecules

with commercially available fullerenes, $C_{60}$, $C_{70}$ and Lu₃N@$I_h$-$C_{80}$ (abbreviated as Lu₃N@$C_{80}$ herein), and as a simple, cheap and sustainable means of solving the problem (Fig. 2). The electronic structure of the fullerene molecules enables π–π interactions between their surface and aromatic organic substituents on the clusters, and their size and spherical shape allows for comfortably cocrystallizing with larger, polyhedral cluster compounds. While cocrystals with $C_{60}$ were previously shown to form with clusters that also crystallize by themselves, like Chevrel-type superatomic clusters[16–18], we demonstrate this technique on the example of amorphous [(PhSn)₄E₆] clusters, the crystal structures of which are otherwise inaccessible. Importantly, in contrast to cocrystals of clusters that crystallize also without fullerenes, and thus do not require specific interactions between the two types of molecules, the π–π interactions are crucial for the cases we report here: Owing to the geometrical relationship between the sizes of the {Sn₄E₆} cores and the distance between the phenyl groups on the cluster surface of [(PhSn)₄E₆], the aromatic substituents of one cluster are not capable of interacting strongly enough with those of neighboring clusters in the pure compound. Hence, the clusters do not assemble in long-range order and the solid remains amorphous. However, via π–π interactions with the six-membered rings of the fullerene molecules, the latter serve as linkers between neighboring cluster units and act as templates to form a crystal lattice. As a proof of a more general applicability of the method, we show that this approach can be expanded to larger fullerenes, specifically $C_{70}$ and Lu₃N@$C_{80}$, which consequently fit better to larger aromatic substituents like naphthyl groups, and thus also enables selective crystallization of amorphous cluster compounds. Quantum-chemical calculations reveal how the crystal formation affects the atomic and electronic structure of the [(PhSn)₄E₆] clusters. The method finally allowed to compare the structural data with previous theoretical predictions[12], and to rationalize and verify the computations.

## Results
### Syntheses and crystal structures
Two intrinsically amorphous adamantane-type cluster compounds, [(PhSn)₄S₆] (**A**) and [(PhSn)₄Se₆] (**B**), were used for our proof-of-principle study to demonstrate the success of the "π-trap" approach. As schematically depicted in Fig. 3a–c, the cocrystallization process

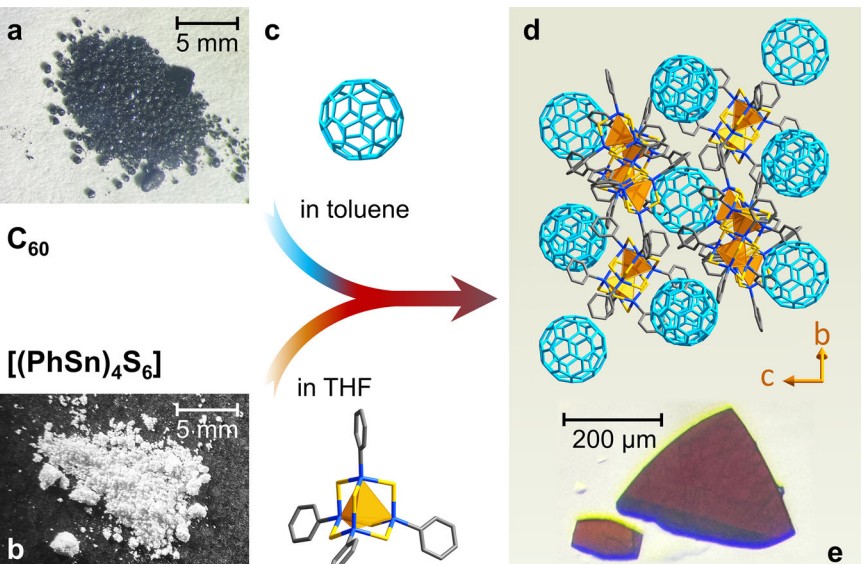

**Fig. 3 | Schematic representation of the preparation procedure for single-crystalline cocrystals of $C_{60}$ and $[(PhSn)_4S_6]$ (A). a** Photograph of microcrystalline $C_{60}$ (as purchased) and structure model of the $C_{60}$ molecule. **b** Photograph of the amorphous powder of pure compound **A** and simplified structure model of the $[(PhSn)_4S_6]$ cluster molecule (see Fig. 1b). **c** Layering of the two solutions comprising the starting materials, $C_{60}$ in toluene and compound **A** in tetrahydrofuran (THF), and diffusion of the solutions into one another. **d** View of the crystal structure of $[(PhSn)_4S_6]_2 \cdot (C_{60}) \cdot (C_7H_8)_{1.2} \cdot (C_4H_8O)_{1.2}$ (**1**) viewed along the crystallographic *a* axis, with cluster pairs visible between the fullerene spheres. **e** Light-microscopic image of single-crystals of compound **1** with scalebar.

with $C_{60}$ involves layering a solution of the cluster compound in THF with a solution of the fullerene in toluene. Slow diffusion of the solutions into each other resulted in the formation of red single crystals at the solution interface after one week (see Supplementary Fig. 1). SCXRD analyses of the single-crystals that were obtained from mixtures involving **A**, unveiled two different products to form simultaneously in the same batch. They feature different ratios of $[(PhSn)_4S_6]$ and $C_{60}$ and also different amounts of crystal solvent molecules ($C_7H_8$ = toluene or $C_4H_8O$ = THF) per formula unit, $[(PhSn)_4S_6]_2 \cdot (C_{60}) \cdot (C_7H_8)_{1.2} \cdot (C_4H_8O)_{1.2}$ (**1**, Fig. 3d, e) and $[(PhSn)_4S_6]_2 \cdot (C_{60})_{1.5} \cdot (C_7H_8)$ (**2**). After adjusting the $C_{60}$:**A** ratio, we were able to obtain both **1** or **2** selectively. Mixtures involving **B** yielded exclusively $[(PhSn)_4Se_6] \cdot (C_{60}) \cdot (C_7H_8) \cdot (C_4H_8O)_{0.5}$ (**3**). Notably, these are the first crystal structures involving notoriously amorphous inorganic clusters compounds in general, and this specific type of amorphous clusters in particular, the mere synthesis and chemical characterization of which was reported as early as in 1987 in the case of **A**[19]. More crystal photographs as well as different views and further details of the crystal structures of the three compounds are provided below and in Supplementary Figs. 2–20.

Figure 4 illustrates the crystal structures of the three cocrystals. The structures show several unique and informative features, which we outline in what follows. A view of the packing schemes in a selected crystallographic direction (see Fig. 4a–c) highlights that all compounds, **1**, **2**, and **3**, comprise pairs of clusters embedded in an environment of $C_{60}$ molecules—albeit in very different, and very complex packing patterns. We observe relatively short distances between the centroids of the hexagonal faces of $C_{60}$ and the clusters' phenyl groups, down to 3.897 Å in compound **1** (Fig. 4a), 3.109 Å in **2**, and 3.250 Å in **3**. This supports our assumption that the crystal lattice is stabilized by face-to-face π–π or C–H·π interactions. Moreover, the shortest distances between chalcogenide atoms (S or Se) and the hexagonal faces of neighboring $C_{60}$ molecules are 3.528 Å for **1**, 3.566 Å for **2**, and 3.710 Å for **3**, indicating another type of secondary interaction.

A close-up of the crystal structure of **1** (Fig. 4d) highlights the π–π interactions between the two types of molecules and, importantly, also within the cluster pairs. When viewed along the Sn···Sn axis, the phenyl groups within the pairs are arranged in staggered positions (Fig. 4e), which allows for a more intense (dominant) interaction of the cluster cores (6.228 Å from center to center) as compared to those between the substituents—as predicted by computational studies of various cluster pairs as minimal model for the interaction[12]. In compound **1** the shortest distance between the centroids of adjacent phenyl groups is measured at 5.568 Å. For comparison, in the analogue crystalline compound $[(PhSi)_4S_6]$ (**C**)[5], the staggered arrangement is different, resulting in a shorter distance of 4.847 Å between adjacent phenyl groups, while the cluster cores are much more distant (7.058 Å, Fig. 4f and Supplementary Fig. 11)—again in excellent agreement with the computations: these indicated stronger substituent···substituent interactions and weaker core···core interactions for the PhSi/S system than for the PhSn/S system (and vice versa)[12].

The core···core distance (center-to-center) between the $[(PhSn)_4E_6]$ clusters in the other pairs are similarly small, 6.284 Å and 6.409 Å (compound **2**), and 6.498 Å (compound **3**), see Supplementary Figs. 12, 13, and 16. As suggested, the predominance of the relatively isotropic core···core interactions in PhSn/S-based clusters surpasses the directional interactions involving the substituents, which is perfectly reflected by the new crystal data, in which the pairs are retained also in the presence of $C_{60}$. This finally rationalizes and explains why Sn-based compounds exhibit a distinctly lower tendency for order as a single compound in the solid state compared to their crystalline PhSi/S-based counterparts.

As another contrast to the ordered situation within crystalline $[(PhSi)_4S_6]$ (**C**)[5], the phenyl groups in compounds **1**, **2**, and **3** show a tendency for disorder (see Fig. 4d, e). This can be explained by the Sn–C bonds being naturally longer than Si–C bonds (2.103(14)–2.126(19) Å for the former versus 1.8540(15)–1.8562(15) Å for the latter, see Supplementary Table 2), which decreases the interaction barrier between the hydrogen atoms of the substituents and the S or Se atoms of the cluster for the heavier homologues. The overall higher flexibility of the phenyl groups obviously weakens their (directed) interaction and thus hampers crystallization. Both cluster cores observed herein, $\{Sn_4S_6\}$ in **1** and **2** and $\{Sn_4Se_6\}$ in **3**, exhibit a notable tendency for molecular distortions. This is particularly obvious in compound **2**, with S–Sn–S angles ranging from 107.84 to 116.03°

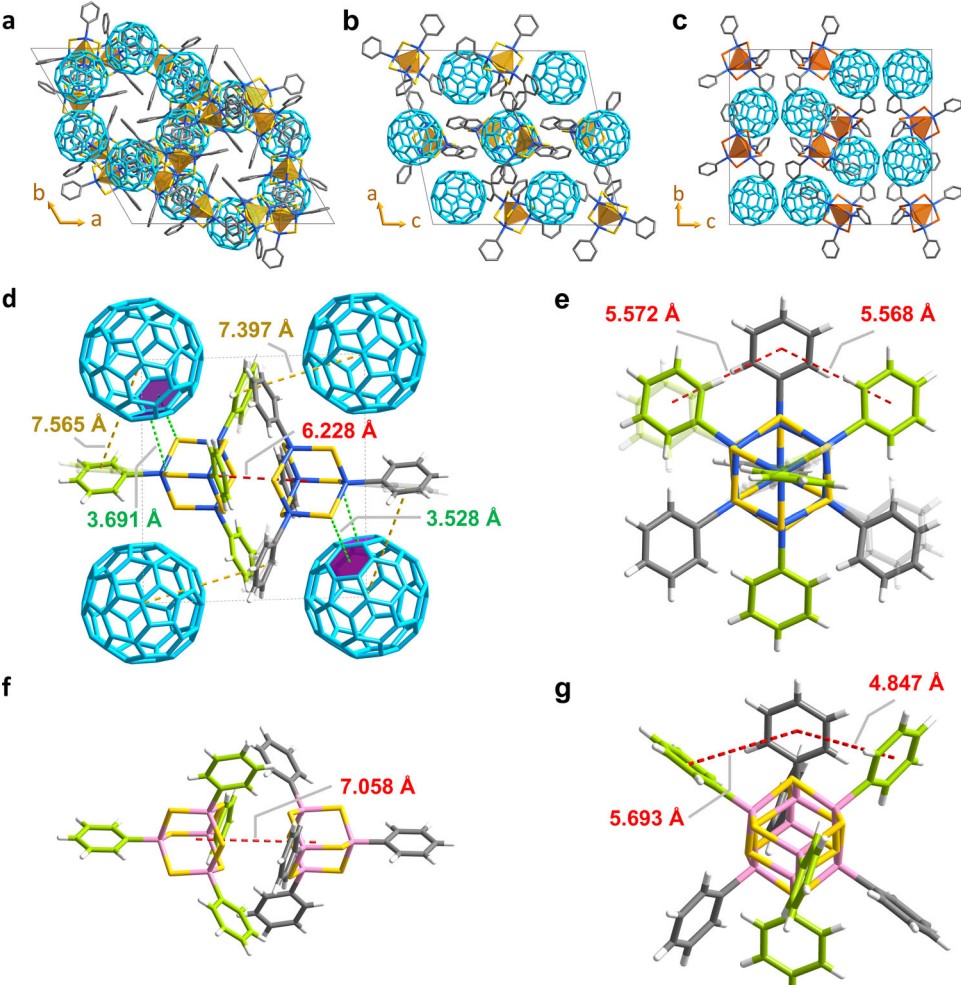

**Fig. 4 | Details of the crystal structures of cocrystals 1, 2, and 3. a** Packing scheme of compound **1**, viewed along the crystallographic *c* axis. **b** Packing scheme of compound **2**, viewed along the crystallographic *b* axis. **c** Packing scheme of compound **3**, viewed along the crystallographic *b* axis. **d** Pair of [(PhSn)$_4$S$_6$] clusters in the crystal structure of **1** interacting with surrounding C$_{60}$ molecules; for clarity, the phenyl groups are depicted in grey and green for the two distinct [(PhSn)$_4$S$_6$] molecules within the pair, respectively (disorder position shown in semitransparent mode). **e** The cluster pair in **1** viewed along the Sn···Sn axis. **f** Pair of [(PhSi)$_4$S$_6$] clusters in the crystal structure of **C**[5], for comparison (Si atoms represented in pink); for clarity, the phenyl groups are depicted in grey and green for the two distinct [(PhSi)$_4$S$_6$] molecules within the pair, respectively. **g** The cluster pair in **C** viewed along the Si···Si axis, for comparison.

(Supplementary Table 2). This range is notably greater than the reported range of S–Si–S angles in [(PhSi)$_4$S$_6$] (**C**; 111.29 to 113.27°)[5], indicating a higher degree of distortion within the Sn/S cluster core than in the Si/S core of the lighter homologue. We suspect that these distortions reflect a certain degree of dynamic behavior, even under crystallization conditions, which refers to increased motion of the atoms as another argument for prohibiting crystallization as a sole compound.

So far, we can summarize that we were able to obtain the first crystallographic structural data from clusters of the [(RT)$_4$E$_6$] family that do not crystallize by themselves. Moreover, these data, as compared to those of the crystalline homologue [(PhSi)$_4$S$_6$] (**C**)[5], served to confirm a theoretical prediction made on the basis of computed pair structures of different clusters in regards of their preference of arranging in ordered structures (crystals) or rather assembling without any long-range periodicity (amorphous powders)[12].

It is worth noting that during the preparation of this manuscript, a similar method known as the "crystalline mate" was showcased in a parallel work[20], but this method was designed for smaller organic molecules only, making it inappropriate for cluster materials. It also required the preparation and provision of specific molecules instead of using commercially available fullerene. Our work was rather inspired

by the cocrystallization of fullerene and [(RCo)$_6$E$_8$] clusters reported recently by the Nuckolls group[16–18]. However, while the Co/E clusters or other species (like cubane or porphyrin molecules)[21–23] do also crystallize without C$_{60}$, this is not the case for the clusters we address in our approach, where addition of C$_{60}$ is the only means of growing crystals comprising those species. As shown by the quoted work though, such cocrystals can achieve new properties through this step. The mere color change upon combination of the colorless (cluster) powder with the black (C$_{60}$) powder to form red crystals also suggests the combination of properties in the case of compounds **1**–**3**. The nature of this combination remained to be explored though.

We therefore investigated whether the intense non-covalent interactions that lead the molecules into an ordered arrangement have an effect on the geometric and electronic structures of the clusters of interest, or whether the latter still behave like the pure substances—especially regarding their macroscopic properties (note that this question does not address WLG, as this requires the amorphous habitus, see above). The visual impression of the crystals, being neither black like C$_{60}$ nor colorless like [(PhSn)$_4$E$_6$] (E = S, Se), but red, could be caused by a mere physical mixture ("blend") of both components, but they may also result from electronic interactions. While the latter would be very interesting in terms of new semiconducting materials,

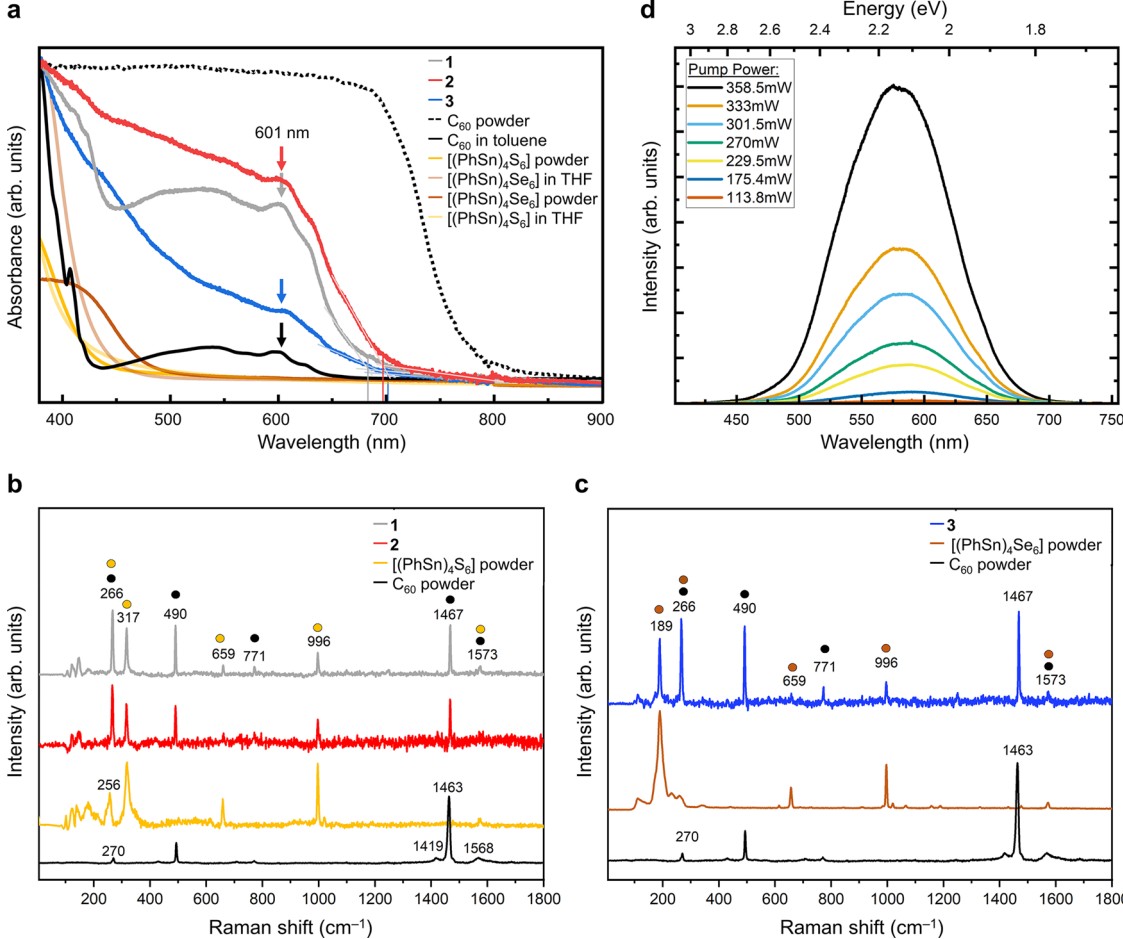

**Fig. 5 | Spectroscopic investigation of parent compounds and cocrystals 1, 2, and 3. a** UV–visible spectra of parent compounds and cocrystals **1**, **2**, and **3**. The spectra were recorded on samples of $[(PhSn)_4S_6]$ and $[(PhSn)_4Se_6]$ as amorphous powders and in solution in tetrahydrofuran (THF), of $C_{60}$ as polycrystalline powder and in toluene, and on single-crystals of **1**, **2**, and **3**. **b** Raman spectra of cocrystals **1** and **2**, compound **A**, and $C_{60}$; the colored dots added above the spectrum of **1** point towards the respective modes in the spectra of the pure substances shown in the respective colors. **c** Raman spectra of cocrystal **3**, compound **B**, and $C_{60}$; the colored dots added above the spectrum of **3** point towards the respective modes in the spectra of the pure substances shown in the respective colors. **d** Nonlinear optical response of the pulverized cocrystal **1** under 1550 nm excitation, as an example for the behavior of such cluster materials upon cocrystallization and pulverization. The fit to the power law for the nonlinearity is shown in Supplementary Fig. 22b.

the first is the primary goal we have in this work, as we aim to get the most reliable and unaffected structural data from this unique class of materials, which requires that they undergo the weakest interaction possible, just enough to order. For judging about this, we investigated the electronic properties of the cocrystals (as well as the separate components for comparison) by means of optical and vibrational spectroscopy and performed concomitant quantum chemical calculations of the crystalline blends. This is detailed in the next two sections.

## Optical absorption and vibrational spectra of 1, 2, and 3 and nonlinear optical response of 1

To gain more insight into the electronic properties of the cocrystals, we studied the optical absorption behavior by means of UV–visible spectra of the pure (amorphous) parent compounds in the solid state and in solution, of $C_{60}$ in the solid state and in solution, and of the single-crystals of **1**, **2**, and **3**, as an example of the compounds presented herein (Fig. 5a). Supplementary Fig. 21 shows Tauc plots that were created using the Tauc formalism $(F(R_\infty)h\nu)^{1/\gamma}$ [24–26], with $\gamma = 1/2$, indicative for a direct allowed optical gap, or $\gamma = 2$, indicative for an indirect allowed optical gap. As an additional proof of the only small impact of $C_{60}$ on the cluster structures of **A** and **B** in the cocrystals, we recorded Raman spectra of compounds **1–3**, **A**, **B**, and $C_{60}$, see Fig. 5b,

c. We also investigated the NLO response of compound **1**, as an example for the effect of cocrystallization. The results are shown in Fig. 5d and Supplementary Fig. 22.

The UV–visible spectra of the cocrystals **1**, **2**, and **3** show an onset of absorption at around 700 nm, slightly red-shifted for compound **3**, followed by **2** and **1**. We assume that the difference between the Sn/Se-based composition of **3** and the Sn/S-based compositions of **1** is not dominant here, but the ratio between the number of $[(PhSn)_4E_6]$ cluster units (E = S, Se) and $C_{60}$ molecules in the cocrystal. In the most red-shifted compound **3**, this ratio is 1:1. In compound **2**, the corresponding ratio is 2:1.5, and **1**, it is 2:1. This is in agreement with the fact that a spectral feature at around 600 nm agrees with the maximum at 601 nm $C_{60}$ in toluene solution. So, for the onset of absorption, we see similar spectroscopic signatures of **1–3** that are controlled by the behavior of $C_{60}$—notably not in the solid, but in solution phase. The more intense absorption that is observed for all cocrystals below 500 nm mostly reflects the absorption characteristics of the pure clusters, for which the solid-state and solution spectra do not differ significantly.

We conclude that in the cocrystal, the two components are widely independent regarding their electronic properties, hence, they behave like a physical mixture ("ordered solid solution"). This would corroborate the fact that the structural data we obtained for the previously

amorphous cluster compounds are those that are inherent to these molecules without great impact from the cocrystal. However, to confirm our working thesis, we performed quantum chemical calculations of the electronic structure of the cocrystals.

The non-invasive and non-destructive nature of the cocrystallization suggested by experimental and DFT-based (see below) optical absorption spectra of the cocrystals was additionally confirmed by Raman spectroscopy. Raman spectra were recorded on single crystals of cocrystals **1**, **2**, and **3**, and compared with the spectra of the pure compounds (see Fig. 5b, c). Two primary modes, the pentagonal pinch C–C stretching vibration mode ($A_g$) at 1463 cm$^{-1}$ and the low-frequency symmetric mode at 490 cm$^{-1}$ associated with C$_{60}$, remain intact in all three cocrystalline samples[27]. The subtle indications of asymmetric stretching ($H_g$) modes (270 and 1568 cm$^{-1}$) of C$_{60}$ in the cocrystals are indicative of a very slight alteration of the polarizability of C$_{60}$ when confined in the cocrystal lattice. Conversely, the −C$_6$H$_5$ ring breathing mode ($A_{1g}$) and C = C stretching mode ($E_{2g}$) observed in the cocrystals at 996 and 1573 cm$^{-1}$, respectively are associated with the ligand coverage from the adamantane-type Sn clusters. Additionally, we have detected Sn-S vibrational bands at range of 100–200, 266, 317 cm$^{-1}$ (for **1**, **2**) and range of 100–200, 189 cm$^{-1}$ (for **3**), which further confirm the structural integrity of the [(PhSn)$_4$S$_6$] and [(PhSn)$_4$Se$_6$] clusters in the cocrystals.

The NLO response of cocrystal **1** was investigated both on single crystals and on a pulverized sample. For excitation of both habitus, we used a 1550 nm continuous-wave infrared (CW-IR) laser diode. Due to the absorption edge of the sample ~700 nm, second-order nonlinear response, i.e., SHG can only be observed for excitation wavelength above 1400 nm. Nevertheless, even at pump densities of 4 kW/cm$^2$ no SHG was observed for both the single crystal and the pulverized sample. In case of the pulverized sample, however, white-light emission was observed (Fig. 5d). The emission spectrum resembles that of the pure (PhSn)$_4$S$_6$, indicating that the original cluster are still intact[4]. An analysis of the input-output characteristics (Supplementary Fig. 22) shows that the intensity of the white-light emission scales proportional to the fourth power of the input intensity. For pure [(PhSn)$_4$S$_6$], the scaling was proportional to the eighth power. We ascribe this discrepancy to the presence of the C$_{60}$ in the pulverized sample.

## First-principles calculations of the electronic properties of the cocrystals 1, 2, and 3

To corroborate and understand the experimental findings, crystals of the compounds **1**, **2**, and **3** have been modeled from first principles. Lattice type and lattice parameters as calculated within DFT-PBE are reported in Supplementary Table 3. The calculated lattice parameters are in very good agreement with the measured values and confirm the experimentally determined structure; upon crystallization into compounds **1**, **2**, and **3**, the cluster cores of **A** and **B** are only very slightly compressed according to the DFT studies (volume changes of 0.2… 0.5%, see Supplementary Table 5). In addition, a comparison of the structural data of **A** and **B** in the gas phase, as free-standing pairs, and in the (calculated and experimentally observed) crystals, see Supplementary Table 6 and Supplementary Fig. 23, indicates a rather small influence of the environment (gas phase or crystal) on bond lengths and angles. Only the phenyl substituents are differently rotated depending on the environment, in line with the previously calculated low rotational energy barriers. This strongly suggests that the C$_{60}$ molecules merely provide a template for the crystallization of the clusters, without profoundly affecting their structures. This is in agreement with the vanishing coupling of the electronic states of C$_{60}$ and of the adamantane-based clusters.

To further investigate the effect of neighboring C$_{60}$ and cluster molecules, we calculated the electrostatic molecular potential for **A** and **B**, both for the free-standing molecules and as extracted from the calculated crystal structures of **1** (as an example of **1** and **2**) and **3**, see Supplementary Fig. 24. This allowed to gather information about their tendency to aggregate in a rather unordered way in the absence of C$_{60}$, but in an ordered way in its presence. The results corroborate that the electrostatic (rather isotropic) core···core interaction between clusters of both **A** or **B** is stronger in the absence of C$_{60}$, which confirms their tendency to aggregate rather arbitrarily instead of crystallizing in ordered structures. Yet, the template provided by the C$_{60}$ molecules allows for the formation of an ordered, periodic structure by means of π–π interactions between the C$_{60}$ surface and the aromatic organic substituents of the clusters.

The formation of the cocrystals leads to the band structures of **1**, **2**, and **3** shown in Fig. 6a–c, which perfectly reflect the mixing of the two compounds. The band structures are very flat as typical for molecular crystals, indicating a minor intermolecular interaction. Moreover, all compounds are semi-conductors, as indicated by an energy gap of about 1.5 eV that separates the top of the valence band from the unoccupied states–in agreement with the experimental findings. Notably, this is significantly smaller than the HOMO-LUMO gap of the parent clusters [(PhSn)$_4$S$_6$] **A** and [(PhSn)$_4$Se$_6$] **B**, calculated by DFT to be 3.199 eV and 2.726 eV. Yet, the original HOMO-LUMO gap can be recognized in the band structure of **1**, **2**, and **3**, as we show in the following on the example of **1**.

Indeed, the spatial localization of the electronic states of the valence band top at −0.3 eV and of the bottom of the second conduction band at about 2.7 eV in Fig. 6a closely resembles the orbital nature of the HOMO and LUMO of **A**; this is in full agreement with the experimental spectrum, in which the signature of the parent clusters can also be identified in smaller wavelength regions. As illustrated in Supplementary Fig. 25, the HOMO is rather localized at the S atoms, while the LUMO is less localized at the S atoms and extended to the substituents. The energetic separation of the electronic states represented in Fig. 6d, e is 2.96 eV, close to the HOMO-LUMO separation in **A**, which further confirms their origin. Thus, the group of localized states that appears within the energy of the HOMO and LUMO of the parent molecules, is more correctly interpreted as a group of localized mid-gap states than as the valence band bottom or conduction band top. In order to explore the origin of these states, we project the density of states (DOS) of **1**, **2**, and **3** onto the different atomic species, see in Fig. 6f. The DOS clearly shows that the localized states have their origin in the carbon atoms of the C$_{60}$ molecules. The wavefunctions associated with these states are indeed localized at the fullerene cages, as shown in Fig. 6g.

Upon crystallization the HOMO and LUMO of the [(PhSn)$_4$S$_6$] and [(PhSn)$_4$Se$_6$] cluster build the conduction and valence band. The fullerene related states lay in between and lead to the top of the conduction band, the bottom of the valence band and mid-gap states. All C$_{60}$-related states are highly localized and almost completely dispersionless. The band structure of **1**, **2**, and **3** thus corresponds to that of the parent compounds, overlaid but not substantially altered by the C$_{60}$ states. The calculated absorbance of the crystals **1**, **2**, and **3** is shown in Supplementary Fig. 27. The spectra substantially resemble the absorption of [(PhSn)$_4$S$_6$] and [(PhSn)$_4$Se$_6$], with an additional signature related to electronic transition involving the C$_{60}$ states (marked by an arrow in the figure). According to the measured data, the onset of the optical absorption is at 700–710 nm for the compounds **2** and **1**, and somewhat redshifted for **3**, due to the smaller bandgap of **B** in comparison with **A**. Interestingly, the C$_{60}$ signatures show a larger scattering than in the measured data, suggesting that the interaction of C$_{60}$ with the clusters is slightly overestimated in the atomistic models.

## Expansion to other systems

As mentioned in the introduction, the method is tailor-made for (1) inherently non-crystalline (amorphous or liquid) compounds that (2) are capable of undergoing π–π interactions—with the aim of obtaining

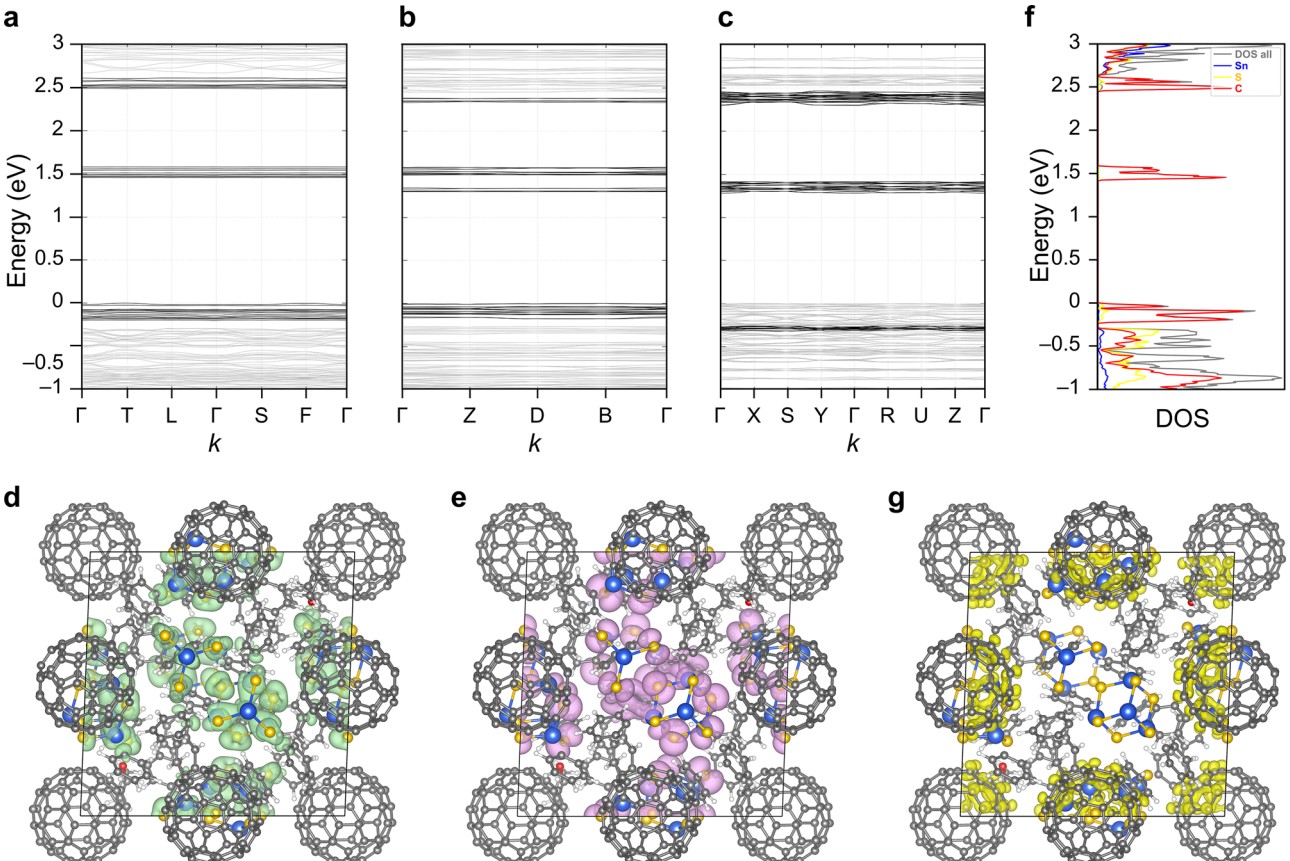

**Fig. 6 | Electronic structures of the compounds calculated within DFT-PBE.**
**a** Electronic band structure for calculated compound **1**. **b** Electronic band structure
for calculated compound **2**. **c** Electronic band structure for calculated compound **3**.
**d** Visualization of the top of the valence band, being strongly localized at the S
atoms, similarly to the lowest unoccupied molecular orbital (LUMO) of **A**.
**e** Visualization of the bottom of the conduction band of **1**, resembling the highest
occupied molecular orbital (HOMO) of **A**. **f** Density of states (DOS) and partial
density of states for compound **1**; the partial density of states is shown in different
colors for different atoms. **g** Visualization of the squared wavefunctions associated
to the mid-gap electronic states in **1**, localized at $C_{60}$. Isosurfaces are drawn at
$0.001\,e\cdot\text{Å}^{-3}$.

precise structural information of rigorously amorphous compounds.
To demonstrate that the method is not restricted to clusters of the
general formula $[(PhSn)_4E_6]$, we expanded the study to further clusters
that exhibit the characteristics (1) and (2) mentioned above. Cocrys-
tallization was also realized with $[(1\text{-}NpSn)_4S_6]$ (**D**; $1\text{-}Np = 1\text{-}$
naphthyl $= C_{10}H_7$), which as a pure compound can only be obtained as
an amorphous powder. Herein, we succeeded in crystallized the clus-
ter with $C_{60}$, but also with larger fullerenes $C_{70}$ and $Lu_3N@C_{80}$. We
obtained single crystals of $[(1\text{-}NpSn)_4S_6]_2\cdot(C_{60})$ (**4**), $[(1\text{-}NpSn)_4S_6]\cdot(C_{70})$
(**5**), and $[(1\text{-}NpSn)_4S_6]_4\cdot(Lu_3N@C_{80})$ (**6**), this way.

SCXRD analyses revealed that compounds **4**–**6** crystallize in the
triclinic space group $P\bar{1}$. Details of the crystal structures are depicted in
Fig. 7. Centroid···centroid distances between the hexagonal faces of
the fullerene molecules and the naphthyl substituents on the clusters
are relatively short, at 3.628 Å for **4**, 3.603 Å for **5**, and 3.719 Å for **6**.
This indicates clearly that the crystal structures are stabilized by face-
to-face π–π or C–H–π interactions. A statistical disorder observed for
compound **6**, with two different orientations of the naphthyl groups
accompanying different positions of one $Lu_3N@C_{80}$ molecule (Sup-
plementary Fig. 20), underlines the importance of the secondary
interactions between the two components in the cocrystals.

$[(1\text{-}NpSn)_4S_6]$ pairs were identified in compounds **4** and **6** (Fig. 7a,
b, e, f) consistent with previous reports, but were absent in compound
**5**, in which the clusters form a one-dimensional chain via π–π inter-
actions of their naphthyl substituents instead (Fig. 7c). This is attri-
buted to the elongated shape of $C_{70}$ as compared to the perfectly
spherical shapes of $C_{60}$ and $C_{80}$, causing the $[(1\text{-}NpSn)_4S_6]$ molecules to

adopt an arrangement more suited to the polar fullerene structure in **5**
(Fig. 7d)[28]. Viewing the cluster pairs in compounds **4** and **6** along the
Sn···Sn axis, the naphthyl groups of the two clusters exhibit staggered
orientations. This goes hand-in-hand with stronger core···core inter-
actions, as indicated by relatively short center-to-center distances of
6.328 Å for **4**, and 6.531 Å for **6**. For comparison, the crystalline com-
pound $[(1\text{-}NpSi)_4S_6]$ (**C**) exhibits a stacked arrangement of the naphthyl
substituents, with a significantly larger core···core distance of 7.439 Å
(Fig. 7g)[12]. In turn, the shortest centroid···centroid distances between
adjacent naphthyl groups are relatively large, 4.937 Å, and 5.001 Å for **4**
and **6**, respectively. This is notably longer than the distance of 3.864 Å
observed in $[(1\text{-}NpSi)_4S_6]$ (Fig. 7h). Altogether, these structural data
serve to prove the prediction of weaker substituent···substituent
interactions and stronger core···core interactions for the $[(1\text{-}NpSn)_4S_6]$
system as compared to $[(1\text{-}NpSi)_4S_6]$ on an atomically-precise experi-
mental basis.

Both, the core···core distances and the substituent···substituent
distances within the $[(1\text{-}NpSn)_4S_6]$ cluster pairs (in **4** and **6**) and the
$[(PhSn)_4S_6]$ cluster pairs (in **1** and **2**) are similar, suggesting similar
interaction strengths. However, the structure of compound **5**, in which
the clusters assemble into chains rather than pairs between the full-
erene molecules, features notably short naphthyl centroid···centroid
distances (3.478–3.573 Å), demonstrating stronger π–π interactions to
be present in the $[(1\text{-}NpSn)_4S_6]$ system. This allows to finally under-
stand the different NLO properties of the two pure compounds, both
of which are X-ray amorphous and were expected to exhibit WLG[5]: the
higher degree of amorphousness in $[(PhSn)_4S_6]$—with no indication of

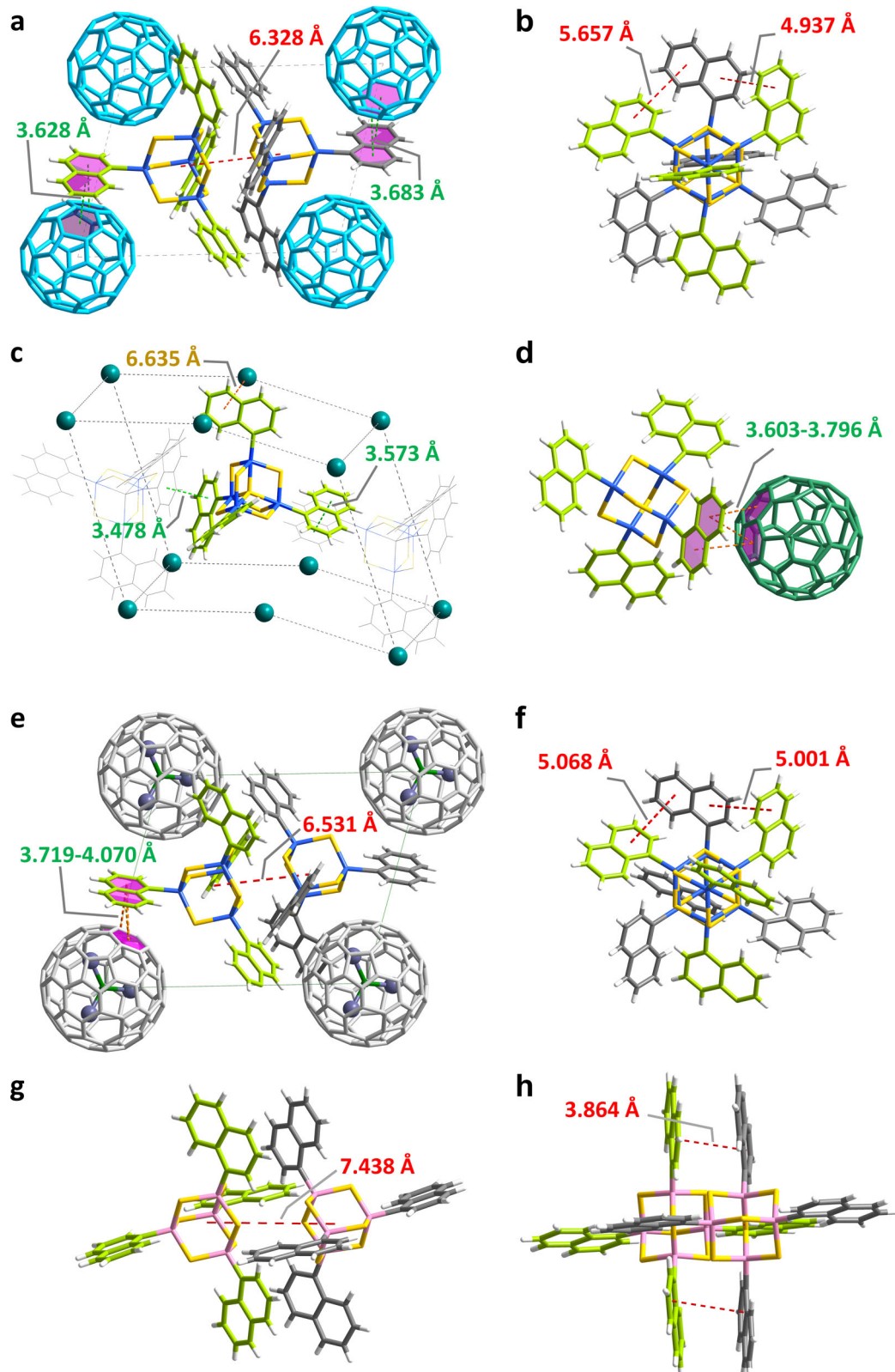

**Fig. 7 | Details of the crystal structures of cocrystals 4, 5, 6, and of compound D[12] for comparison. a** Pair of [(1-NpSn)$_4$S$_6$] clusters in the crystal structure of **4** interacting with surrounding C$_{60}$ molecules. **b** The cluster pair in **4** viewed along the Sn···Sn axis. **c** Position of a [(1-NpSn)$_4$S$_6$] cluster within a chain of clusters and surrounded by C$_{70}$ molecules (simplified as green spheres) in the crystal structure of **5**. **d** [(1-NpSn)$_4$S$_6$] cluster in **5** interacting with adjacent C$_{70}$ molecules. **e** [(1-

NpSn)$_4$S$_6$] clusters in the crystal structure of **6** interacting with surrounding Lu$_3$N@C$_{80}$ molecules. **f** The cluster pair in **6** viewed along the Sn···Sn axis. **g** Pair of [(1-NpSi)$_4$S$_6$] clusters in the crystal structure of **D** for comparison (Si atoms represented in pink). **h** The cluster pair in **D** viewed along the Si···Si axis for comparison. For clarity, the naphthyl groups are depicted in grey and green for the two distinct [(1-NpSi)$_4$S$_6$] molecules within the pair, respectively.

any order beyond the molecular range—results in WLG, while the larger (supramolecular and thus medium-range) order of $[(1\text{-NpSn})_4S_6]$ clusters in the corresponding powder supports SHG.

We note in addition that the "π-trap" concept also offers selectivity. While $[(1\text{-NpSn})_4S_6]$ clusters exhibiting naphthyl groups on their surfaces can be crystallized with all three fullerenes tested in the cocrystallization processes, clusters with R being phenyl do exclusively form cocrystals with $C_{60}$. The application of $C_{70}$ and $Lu_3N@C_{80}$ can thus be used as a means of selective trapping of one cluster type from a mixture of clusters. We attribute this selectivity to two effects: firstly, to the larger steric demand of clusters with Np substituents fitting better to the larger size of the $C_{70}$ and $Lu_3N@C_{80}$ molecules and thus facilitating the packing of both components in the cocrystal. Second, and more importantly, to the surfaces of $C_{70}$ and $Lu_3N@C_{80}$ that, with their smaller curvature, provide of a more suitable counterpart for the more extended π system of the naphthyl substituents.

## Discussion

Overall, through our "π-trap" approach, we demonstrate that inherently amorphous adamantane-like cluster can be trapped in a crystal structure by π–π interactions with surrounding $C_{60}$, $C_{70}$, or $Lu_3N@C_{80}$ molecules, which we showcased for the important white-light-emitting compounds $[(PhSn)_4S_6]$, $[(PhSn)_4Se_6]$, and $[(1\text{-NpSn})_4S_6]$.

The structural data obtained through this approach confirmed preliminary hypotheses regarding the structural features of these clusters, which were previously made based on theoretical predictions, by providing critical evidence of the diverse molecular interactions within these systems. We demonstrated that π–π interactions between clusters and fullerene molecules serve to overcome distortions of the clusters and the predominance of core⋯core interactions of Sn-based clusters, which were previously identified as reasons for such clusters to remain inevitably amorphous. Moreover, the findings revealed important differences in the assembly behavior of clusters with Ph substituents and Np substituents, which helped to understand why the two pure systems $[(PhSn)_4S_6]$ and $[(1\text{-NpSn})_4S_6]$, as X-ray amorphous solids, show different NLO responses.

By a combined experimental and theoretical study, we demonstrated that the cocrystals are more than a mere source of structural information, but exhibit a combination of the electronic properties of the two underlaying components. The enhanced understanding of the structural and interactive properties of the amorphous clusters contributes to the ability to predict, design, and control their features, paving the way for future advancements in the targeted development of these compounds.

## Methods

### General experimental methods

All solvents were dried by standard methods and were commercially obtained. All reaction and sample preparations were carried out under inert atmospheres using standard Schlenk techniques or in an argon-filled glovebox. $C_{60}$ (98%, Sigma-Aldrich Chemie GmbH, US), $C_{70}$ (98%, Sigma-Aldrich Chemie GmbH, US), $Lu_3N@C_{80}$ (97%, SES Research Inc., US), $PhSnCl_3$ (98%, Sigma-Aldrich Chemie GmbH, US), $Ph_4Sn$ (95%, abcr GmbH, Germany), $NH_3$ (99.98 %, Air Liquide Deutschland GmbH, Germany), Na (Sigma-Aldrich Chemie GmbH), S (99.5 %, 100 mesh, Thermo Scientific Chemicals) and Se (99.5%, 200 mesh, Thermo Scientific Chemicals), Mg (99.5%, Sigma-Aldrich), Bromonaphthaline (97%, abcr), Celite™ 545 (150 mesh, Thermo Fischer Scientific), $Na_2SO_4$ (99.5%, Thermo Fischer Scientific), and ethylacetate (99.9%, VWR International) were used as received. Chlorotrimethylsilane (98% Thermo Fischer Scientific) and $SnCl_4$ (98% Thermo Fischer Scientific) were freshly distilled prior to use. Solvents THF, toluene, $n$-hexane, n-pentane, and diethylether were dried over sodium and freshly distilled onto molecular sieve prior to use. Dichloromethane were dried over molecular sieve and freshly distilled onto molecular sieve prior to use.

$[(PhSn)_4S_6]$ (**A**), $[(PhSn)_4Se_6]$ (**B**), and $[(1\text{-NpSn})_4S_6]$ (**D**) were prepared according to literature procedures[7]:

### $[(PhSn)_4S_6]$ (A)

To an ice-cold solution of bis(trimethylsilyl)sulfide (3.1 mL, 14.49 mmol, 1.6 eq.) in 20 mL of toluene, phenyltintrichloride (1.5 mL, 9.12 mmol, 1 eq.) was added dropwise. The solution was stirred for 12 h. A colorless precipitate formed that was filtered off and washed thrice with 5 mL of $n$-hexane. Residual volatiles were removed under reduced pressure. $[(PhSn)_4S_6]$ (**A**) was obtained as a colorless, fine powder with a yield of 1.62 g (73.2 %). $^{119}Sn$ {$^1H$} NMR (186.5 MHz, $CD_2Cl_2$): 83.4 ppm.

### $[(PhSn)_4Se_6]$ (B)

To an ice-cold solution of bis(trimethylsilyl)selenide (2.4 mL, 9.57 mmol, 1.58 eq.) in 20 mL of toluene, phenyltin trichloride (1.0 mL, 6.06 mmol, 1 eq.) was added dropwise. The solution was stirred for 16 h. A yellow precipitate formed that was filtered off and washed thrice with 5 mL of $n$-hexane. Residual volatiles were removed under reduced pressure. $[(PhSn)_4Se_6]$ (**B**) was obtained as a pale-yellow powder with a yield of 1.53 g (80.2 %). $^{119}Sn$ {$^1H$} NMR (112 MHz, $CD_2Cl_2$): −97.8 ppm.

### $[(1\text{-NpSn})_4S_6]$ (D)

To an ice-cold solution of bis(trimethylsilyl)sulfide (3.1 mL, 14.49 mmol, 1.6 eq.) in 20 mL of toluene, 1-naphthyltintrichloride (3.21 mg, 9.12 mmol, 1 eq.) was added dropwise. The solution was stirred for 12 h. A colorless precipitate formed that was filtered off and washed thrice with 5 mL of $n$-hexane. Residual volatiles were removed under reduced pressure. $[(1\text{-NpSn})_4S_6]$ (**D**) was obtained as a colorless, fine powder with a yield of 1.62 g (73.2 %). $^{119}Sn$ {$^1H$} NMR (186.5 MHz, $CD_2Cl_2$): 83.4 ppm.

### Bis(trimethylsilyl)sulfide or bis(trimethylsilyl)selenide

Sodium (21.52 g, 0.936 mol or 6.53 g, 0.284 mol) was cut and put in a three necked 1 L round bottom flask. The flask was cooled to −78 °C, before ammonia (~500 mL) was condensed onto the sodium and mechanically stirred. The deep-blue solution was warmed to about −45 °C before adding sulfur (15.38 g, 0.479 mol) or selenium (11.78 g, 0.150 mol) portion-wise over a time-span of 8 h. The addition was stopped as soon as the suspension persisted to be colorless. Throughout the addition, the temperature was kept between −38 °C and −48 °C. The suspension was stirred for another 2 h or 12 h in the cooling bath. Subsequently, the ammonia was evaporated over 16 h, whereupon the remaining solid was dried in vacuo for 16 h. THF (500 mL) was added to the solid, and the suspension was stirred in an ice bath, while chlorotrimethylsilane (110 mL, 0.865 g/cm³, 0.867 mol or 36 mL, 0.865 g/cm³, 0.284 mol) was added dropwise. The mixture was stirred for another 16 h in the ice bath, and then filtered. The filtrate was first distilled at normal pressure to remove the THF. Bis(trimethylsilyl)sulfide was collected at reduced pressure at a head temperature of 82 °C (50–60 Torr) as a colorless liquid with a yield of 49.2 g (63.7%). $^1H$ NMR (300 MHz, $CDCl_3$): 0.34 ppm. Bis(trimethylsilyl) selenide was isolated by vacuum distillation as a slightly yellow liquid with a yield of 36.3 g (56.7%). $^1H$ NMR (300 MHz, $CDCl_3$): 0.20 ppm.

### 1-Naphthyltintrichloride

[29]Tetra-1-naphthyltin (3.00 g, 4.78 mmol, 1 eq.) was combined with 1.67 mL of $SnCl_4$ (3.73 g, 14.3 mmol, 3 eq.) in a Schlenk flask furnished with a reflux condenser. The mixture was heated up to 150–190 °C using an oil bath and stirred for 2 h for complete conversion. Excess $SnCl_4$ was removed under reduced pressure to obtain a dark brown liquid. The black solid was taken up in 50 mL of dichloromethane. The suspension was filtered through Celite™, and the solvent was evaporated under reduced pressure. The remaining colorless solid was recrystallized from toluene to afford $1\text{-NpSnCl}_3$ as colorless crystals.

Yield: 51% (3.43 g, 9.73 mmol). M.p., 71–73 °C. Anal. calcd. for $C_{10}H_7Cl_3Sn$: C, 34.10; H, 2.00. Found: C, 33.99; H, 1.87. $^1H$ NMR ($C_6D_6$, 300 MHz): δ 7.97–7.89 (dd, 1H, H6), 7.50–7.30 (m, 3H, H2, H4, H5), 7.12–7.03 (m, 2H, H7, H8), 6.91–6.83 (m, 1H, H3) ppm. $^{13}C$ NMR ($C_6D_6$, 75.5 MHz): δ 136.1 ($^1J(^{13}C-^{119}Sn) = 1098$ Hz, $^1J(^{13}C-^{117}Sn) = 1050$ Hz, C1), 135.1 ($^2J(^{13}C-^{119}Sn) = 85.5$ Hz, $^2J(^{13}C-^{117}Sn) = 81.7$ Hz, C8a), 134.8 ($^3J(^{13}C-^{119}Sn) = 103$ Hz, $^3J(^{13}C-^{117}Sn) = 98.3$ Hz, C4a), 134.7 ($^2J(^{13}C-^{119/117}Sn) = 63.4$ Hz, C2), 133.3 ($^4J(^{13}C-^{119/117}Sn) = 27.5$ Hz, C4), 129.4 ($^4J(^{13}C-^{119/117}Sn) = 20.7$ Hz, C5), 128.7 ($^4J(^{13}C-^{119/117}Sn) = 6.9$ Hz, C7), 125.5 ($^3J(^{13}C-^{119/117}Sn) = 58.3$ Hz, C8), 127.3 (C6), 125.8 ($^3J(^{13}C-^{119}Sn) = 141$ Hz, $^3J(^{13}C-^{117}Sn) = 135$ Hz, C3) ppm. $^{119}Sn$ NMR ($C_6D_6$, 112 MHz): δ −62.3 ppm.

## Tetra(1-naphthyl)tin

[29]Magnesium (17.0 g, 700 mmol, 7 eq.) were covered by 700 mL of THF, and 1-bromonaphthaline (83.4 mL, 600 mmol, 6 eq.) in 150 ml THF was added to form a Grignard solution. After complete addition, the reaction was refluxed for 2 h. A second flask equipped with a mechanical stirrer and a reflux condenser was charged with $SnCl_4$ (11.7 mL, 100 mmol, 1 eq.) in 1000 mL of THF and cooled with an ice bath. The Grignard solution was then transferred through a cannula to the $SnCl_4$ solution whilst hot in order to avoid precipitation of the Grignard reagent. The resulting mixture was refluxed for 2 h and stirred for 12 h at room temperature. Then, the solution was filtered through Celite™ and the solvent was evaporated under reduced pressure. Water (1000 mL) was added to the resulting residue, and the mixture was then extracted with 250 mL of dichloromethane. The organic phase was dried over $Na_2SO_4$, filtered, and the solvent was evaporated under reduced pressure. The product was suspended in 250 mL of diethylether, filtered, and washed with 250 mL of diethylether and subsequently with 250 mL of n-pentane. The product was dried in an oven at 110 °C for 12 h. For structural analysis of 1-Np$_4$Sn, a small amount was recrystallized from ethylacetate to obtain colorless crystals. Yield: 72% (45.2 g, 72.0 mmol). M.p., 230-232 °C. Anal. calcd. for $C_{20}H_{14}Sn$: C, 76.58; H, 4.50. Found: C, 75.36; H, 4.40. $^1H$ NMR ($C_6D_6$, 300 MHz): δ 8.33 (d, 4H, $^3J(H4-H3) = 8.3$ Hz, H4), 8.12 (d, 4H, $^3J(H2-H3) = 6.6$ Hz, H2), 7.62 (d, 4H, $^3J(H8-H7) = 8.2$ Hz, H8), 7.54 (d, 4H, $^3J(H5-H6) = 8.1$ Hz, H5), 7.12–7.00 (m, 8H, H6, H7), 6.83 (dd, 2H, H3) ppm. $^{13}C$ NMR ($C_6D_6$, 75.5 MHz): δ 140.7 ($^1J(^{13}C-^{119}Sn) = 520$ Hz, $^1J(^{13}C-^{117}Sn) = 497$ Hz, C1), 139.4 ($^2J(^{13}C-^{119}Sn) = 34.7$ Hz, $^2J(^{13}C-^{117}Sn) = 33.8$ Hz, C8a), 137.6 ($^2J(^{13}C-^{119/117}Sn) = 38.1$ Hz, C2), 134.5 ($^3J(^{13}C-^{119}Sn) = 37.6$ Hz, $^3J(^{13}C-^{117}Sn) = 36.4$ Hz, C4a), 130.6 ($^4J(^{13}C-^{119/117}Sn) = 32.2$ Hz, C4), 130.3 ($^3J(^{13}C-^{119/117}Sn) = 11.7$ Hz, C8), 129.3 ($^3J(^{13}C-^{119/117}Sn) = 43.8$ Hz, C3), 126.4 (C7), 126.1 (C6) ppm. $^{119}Sn$ NMR ($C_6D_6$, 112 MHz): δ −118.8 ppm.

## Synthesis of $[(PhSn)_4S_6]_2 \cdot (C_{60}) \cdot (C_7H_8)_{1.2} \cdot [(C_4H_8O)]_{1.2}$ (1)

$[(PhSn)_4S_6]$ (9.7 mg, 0.010 mmol) and $C_{60}$ (5 mg, 0.006 mmol) were dissolved in THF (3 mL) and toluene (3 mL), respectively, and filtered through 0.8 μm PTFE filters. The solution of $C_{60}$ was slowly layered on the top of the $[(PhSn)_4S_6]$ solution. The vial was placed in a freezer at −19 °C for one week, after which the formation of red cuboidal 1 single crystals of 1 could be observed (yield: 13.8% based on $[(PhSn)_4S_6]$).

## Synthesis of $[(PhSn)_4S_6]_2 \cdot (C_{60})_{1.5} \cdot (C_7H_8)$ (2)

$[(PhSn)_4S_6]$ (9.7 mg, 0.010 mmol) and $C_{60}$ (14.0 mg, 0.019 mmol) were dissolved in THF (3 mL) and toluene (5 mL), respectively, and filtered through 0.8 μm PTFE filters. The solution of $C_{60}$ was slowly layered on the top of the $[(PhSn)_4S_6]$ solution. The vial was placed in a freezer at −19 °C for one week, after which the formation of red cuboidal single crystals of 2 could be observed (yield: 6.4% based on $[(PhSn)_4S_6]$).

## Synthesis of $[(PhSn)_4Se_6] \cdot (C_{60}) \cdot (C_7H_8) \cdot (C_4H_8O)_{0.5}$ (3)

Compound 3 was obtained using procedures similar to those for 1·2toluene: $[(PhSn)_4Se_6]$ (0.010 mmol) and $C_{60}$ (0.010 mmol) were dissolved in THF (3 mL) and toluene (3 mL), respectively, and filtered through 0.8 μm PTFE filters. The solution of the fullerene was slowly layered on top of the tin cluster solution, and the vessel was placed in a freezer at −19 °C. After one week, a black amorphous precipitate could be observed in the purple solution. After filtration through 0.8 μm PTFE filters, the purple filtrate was transferred into an 8 mL vial, and the solvent was evaporated slowly at room temperature over a period of one more week. Red single crystals formed on the walls of the vial (yield: 9.4% based on $[(PhSn)_4Se_6]$).

## Synthesis of $[(1\text{-}NpSn)_4S_6]_2 \cdot [C_{60}]$ (4)

$[(1\text{-}NpSn)_4S_6]$ (11.7 mg, 0.010 mmol) and $C_{60}$ (4.0 mg, 0.006 mmol) were dissolved in THF (3 mL) and toluene (3 mL), respectively, and filtered through 0.8 μm PTFE filters. The solution of $C_{60}$ was slowly layered on top of the $[(1\text{-}NpSn)_4S_6]$ solution and the vessel was placed at room temperature, and evaporated slowly over a period of one week. Red single crystal formed on the walls of the vial (yield: 9.1% based on $[(1\text{-}NpSn)_4S_6]$).

## Synthesis of $[(1\text{-}NpSn)_4S_4] \cdot [C_{70}] \cdot (C_7H_8)$ (5)

In a similar procedure as applied for the synthesis of compound 4, $[(1\text{-}NpSn)_4S_6]$ (0.010 mmol) and $[C_{70}]$ (0.010 mmol) were dissolved in THF (3 mL) and toluene (3 mL), respectively, and filtered through 0.8 μm PTFE filters. The solution of the fullerene was slowly layered on top of the tin cluster solution, and the vessel was placed in a freezer at −19 °C. After two weeks, a black amorphous precipitate could be observed in the purple solution. After filtration through 0.8 μm PTFE filters, the purple filtrate was transferred into an 8 mL vial, and the solvent was evaporated slowly at room temperature over a period of one week. Red single crystals formed on the walls of the vial (yield: 7.2% based on $[(1\text{-}NpSn)_4S_6]$).

## Synthesis of $[(1\text{-}NpSn)_4S_6]_4 \cdot [Lu_3N@C_{80}]$ (6)

In a similar procedure as applied for the synthesis of compounds 4, and equivalent to the synthesis of 5, $[(1\text{-}NpSn)_4S_6]$ (0.010 mmol) and $[Lu_3N@C_{80}]$ (0.010 mmol) were dissolved in THF (3 mL) and toluene (3 mL), respectively, and filtered through 0.8 μm PTFE filters. The solution of the fullerene was slowly layered on top of the tin cluster solution, and the vessel was placed in a freezer at −19 °C. After two weeks, a black amorphous precipitate could be observed in the purple solution. After filtration through 0.8 μm PTFE filters, the purple filtrate was transferred into an 8 mL vial, and the solvent was evaporated slowly at room temperature over a period of one week. Red single crystals formed on the walls of the vial (yield: 6.0% based on $[(1\text{-}NpSn)_4S_6]$).

## Single crystal X-ray diffraction

The data for the X-ray structural analyses was collected at T = 150.0 or 180 K on a STOE STADI VARII diffractometer with Ga/Kα radiation (λ = 1.34143 Å) for all compounds. Data collection, integration, scaling (ABSPACK), and absorption correction were performed in X-area. The structures were solved using SHELXT from SHELXL-2018/136, and refined by full matrix least-squares methods against $F^2$ with the SHELXL program[30]. Olex2 was used for viewing and to prepare CIF files[31]. Refinement was performed with anisotropic temperature factors for all non-hydrogen atoms. Hydrogen atoms were calculated on idealized positions. Figures were created with Diamond. Owing to heavy disorder of the atoms of the solvent (toluene and THF) molecules of compound 1, these were retracted by applying a solvent mask in Olex2 in order to get an optimal model of the cluster and $C_{60}$ molecules. The retracted electron density correlates to 0.6 equivalents of toluene and 0.6 equivalents of THF in the asymmetric unit.

## Optical absorption spectroscopy

UV–visible spectra of 1, 2, and 3 single crystals, as well as $C_{60}$, $[(PhSn)_4S_6]$, and $[(PhSn)_4Se_6]$ powders, were measured in absorbance mode using a Varian Cary 5000 UV/VIS/NIR spectrometer from Agilent, equipped with a Praying Mantis accessory for solid-state samples.

Additionally, $C_{60}$ in toluene solution was measured in absorbance mode using the same spectrometer. Tauc plots were generated according to the Tauc formalism[24–26], $(F(R_\infty)h\nu)^{1/\gamma}$, with $\gamma = \frac{1}{2}$, indicative for a direct allowed optical gap, and $\gamma = 2$, indicative for an indirect allowed optical gap direct band gap.

## Raman spectroscopy

Raman spectra were measured using a Renishaw inVia confocal Raman spectrometer. The instrument is equipped with 532 nm and 785 nm diode-pump solid-state laser source, monochromator, diffraction gratings, and an ultra-high sensitivity CCD detector. Objective lenses with 100× magnification were used to focus the incident beam towards the crystals. Data acquisition was performed using single crystals at room temperature using the WiRE software, with the detector providing a spectral resolution of 0.5 cm⁻¹. All measurements were conducted using a 5 mW laser power with a 10-s acquisition time. Spectral acquisition for $C_{60}$ was performed using a 532 nm laser, while [(PhSn)$_4$S$_6$], [(PhSn)$_4$Se$_6$], and cocrystalline samples were measured using a 785 nm laser.

## Measurement of the nonlinear optical response

The NLO response was measured in an in-house build setup shown in Supplementary Fig. 22a. For excitation of the sample a continuous-wave 1550 nm diode laser (CNI-MDL-N-1550-2500-FC) is used. The laser is focused onto the sample using an $f = 40$ mm lens. The samples are placed on a cover glass slip and mounted in a custom build vacuum chamber. Before the excitation the chamber is evacuated to at least $2\cdot10^{-6}$ mbar. The light emitted by the sample is collected in transmission geometry. First collimated using an $f = 100$ mm lens and then focused using an $f = 40$ mm lens. For detection a compact CCD spectrometer (OceanOptics HR2000) is used. The spectral response of the whole detection system has been corrected using the emission of a standard tungsten-halogen emitter at known temperature (OceanOptics HL-3P).

## Quantum chemical studies

Isolated molecular clusters as well as related crystals are modeled within Desnsity Fuctional Theory (DFT) and periodic boundary conditions. Thereby, we employed the Vienna ab initio simulation package[32,33] to evaluate the structural, electronic, and optical properties of the investigated compounds. Molecular clusters in the gas phase were modeled within the molecule-in-a-box approach. Cubic boxes with a volume of 51179.6 Å³, explicitly designed to decouple periodic images of the clusters, were employed. Crystalline solids were modeled within their unit cell. The atomic positions were optimized until the Hellmann–Feynman forces acting on each atom are lower than 0.001 eV·Å⁻¹ [34]. The ion-electron interaction was described by projected augmented wave pseudopotentials[35,36], implementing the PBE formulation[37,38] of the generalized gradient approximation[39]. The van der Waals (vdW) DFT-D3 method with Becke-Johnson damping[40,41] was applied in all calculations to account for dispersion forces, as (semi)local exchange-correlation (XC) functionals do not properly describe the long-range vdW interactions. Plane waves up to a cutoff of 600 eV were used as the basis for the expansion of electron wave functions. Due to the large cell size, gamma-point calculations were performed. The imaginary part of the frequency-dependent dielectric tensor $\varepsilon_{\alpha\beta}^i$, $\alpha, \beta = x, y, z$, was calculated as the sum over empty states at each point of the first Brillouin zone[42]. The real part of the dielectric tensor $\varepsilon^r$ was obtained by Kramers–Kronig transformations. The absorbance coefficient $k(\omega)$ was calculated from the dielectric function as given in Eq. (1), as

$$k_j(\omega) = \frac{1}{\sqrt{2}} \sqrt{\sqrt{\left(\varepsilon_{jj}^r(\omega)\right)^2 + \left(\varepsilon_{jj}^i(\omega)\right)^2} - \varepsilon_{jj}^r(\omega)} \tag{1}$$

where $\varepsilon^r(\omega)$ and $\varepsilon^i(\omega)$ are the real part and the imaginary part of the dielectric tensor, respectively.

## Data availability

The structures of compounds **1**–**6** were determined by single-crystal X-ray diffraction. Crystallographic data for the structure reported in this Article have been deposited at the Cambridge Crystallographic Data Centre, under deposition numbers CCDC-2419991 (**1**). CCDC-2419992 (**2**), and CCDC-2419993 (**3**), CCDC-2456507 (**4**), CCDC-2456508 (**5**), CCDC-2456509 (**6**). A copy of the data can be obtained free of charge via https://www.ccdc.cam.ac.uk/structures/. The Cartesian coordinates of all optimized structures and the respective SCF energies generated in this study are provided in the Supplementary Information, summarized in "Source Data.zip". The files comprise all necessary data for reproducing the values. All non-default parameters for the computational studies are given in the Supplementary Information together with the corresponding references of the used methods. All data are also available from the corresponding author upon request. Source data are provided with this paper.

## Code availability

The Vienna Ab initio Simulation Package (VASP) is a computer program for atomic scale materials modelling from first principles (see also refs. 32,33). The copyright-protected software is property of the VASP Software GmbH and available at https://www.vasp.at upon license purchase from the VASP Software GmbH or from an official reseller. The VASP version 6.4.2 and the official PAW potentials (see also refs. 35,36) are employed for the calculations presented in this work.

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

## Acknowledgements

This work is supported by the German Research Foundation (DFG) through the research group FOR2824 (Grant No. 398143140). Calculations for this research were conducted on the Lichtenberg high performance computer of the TU Darmstadt and at the Höchstleistungrechenzentrum Stuttgart (HLRS). The authors furthermore acknowledge the computational resources provided by the HPC Core Facility and the HRZ of the Justus-Liebig-Universität Gießen. Y.W. thanks Dr. Zhou Wu and Katrin Beuthert (Karlsruhe Institute of Technology, Institute of Nanotechnology) for their help with the SC-XRD measurements.

## Author contributions

J.C. synthesized the compounds $[(PhSn)_4E_6]$ (E = S and Se), S.N. synthesized the compound $[(1-NpSn)_4S_6]$. Y.W. conceived and performed the cocrystallization experiments, collected single-crystal X-ray crystallographic data, solved and refined the structures. A.J. performed the Raman measurements and N.W.R. investigated the nonlinear optical response. K.E. and F.Z. performed the calculations and analysis with support from S.S.; S.D. and N.R. conceptualized the study and supervised the experimental work. The manuscript was written through contributions of all authors, and all authors reviewed the manuscript. All authors have given approval to the final version of the manuscript.

## Funding

## Competing interests

The authors declare no competing interests.
