## [Transparent Peer Review file · Nature Communications]

The π -Trap Approach for Obtaining Crystal Structure Data of Inherently Amorphous Cluster Compounds

Corresponding Author: Professor Stefanie Dehnen

Version 0:

Reviewer comments:

Reviewer #1

(Remarks to the Author)

This manuscript by Dehnen and co-authors introduces a novel approach to determining the precise crystal structures of inherently amorphous cluster compounds. Their method overcomes the size limitations of traditional techniques by using π - π interactions between organic ligands on the cluster molecules and the faces of C₆₀ molecules, effectively forcing the clusters into an ordered co-crystal lattice. Using this strategy, the authors not only resolve the precise crystal structures of amorphous cluster compounds of the type [(PhSn)₄E₆] (E = S, Se) but also elucidate why these clusters do not crystallize on their own. The commercial availability of C₆₀ further facilitates the adoption of this method. This discovery provides valuable insights that may contribute to future advancements in the design and development of amorphous compounds. I recommend its publication in Nature Communications following minor revisions.

1. The authors' group has studied numerous adamantane-like clusters. I'm curious—were the structures of both adamantane-based clusters in this manuscript previously calculated? Additionally, have the authors compared your crystal structures with earlier theoretical models?
2. Have the authors investigated the generation of white light in these crystals?
3. Figure 2 looks great, but the designation of the green cubic structure as "MOF" is inaccurate. In fact, it represents only the secondary building units (SBUs) rather than the full metal-organic framework. The authors should revise the labeling for precision.
4. In Figure 3, the authors use simplified grey spheres to represent phenyl groups. However, this depiction makes them resemble methyl groups, which may lead to misinterpretation. Additionally, the π - π interactions are not explicitly illustrated beyond this representation. It is suggested to retain the explicit phenyl group structure to ensure clarity and accuracy.
5. In Figure 5, the onset of absorption for compounds 1, 2, and 3 is not clearly distinguishable. Although the authors have marked certain lines, the visibility could be improved. It is recommended to use higher-weight lines or insert a zoomed-in inset to enhance clarity.
6. The proposed approach would be more compelling if the authors applied it to other types of clusters. Furthermore, it would be worthwhile to explore whether other fullerenes (like C₇₀, Sc₃N@C₈₀) can be incorporated using the same methodology.
7. Typos in the manuscript:
 - 1) Page 3: "for the WLG phenomenon, and is has also not been clarified until today"
 - 2) Page 5, the caption of Figure 3: "for a single-crystalline cocrystals of C60 and [(PhSn)4S6]"

Reviewer #2

(Remarks to the Author)

In this manuscript, the authors propose the ' π -trap' method as an unrestricted crystal sponge for crystallizing inherently amorphous cluster compounds. However, I have several concerns that need to be addressed before this manuscript can be considered suitable for publication in Nature Communications.

1. Misapplication of the 'Crystal Sponge' Concept & Conceptual Clarity:

The manuscript refers to their method as an "unrestricted crystal sponge," which is a misapplication of the well-established concept of "crystal sponge" based on metal-organic frameworks (MOFs), as pioneered by Fujita et al. In contrast to MOF-based systems, the ' π -trap' method involving C₆₀ is fundamentally different in both structure and mechanism. This could confuse readers and should be clarified. The authors must revise their nomenclature to more accurately describe the methodology and distinguish it from MOF-based crystal sponge approaches. Additionally, the manuscript would benefit from

a more detailed comparison between the 'π-trap' and conventional co-crystallization methods that use C60, as this distinction is not clearly delineated. The manuscript should also explore the differences in crystallization mechanisms to justify how the π-π interactions in the proposed method are distinct from typical co-crystallization strategies.

2. Lack of Evidence Supporting the Mechanism of Crystallization:

The authors claim that the assembly process is induced by the target molecular skeleton, but they provide no direct evidence for this assertion. Given that C60 has a well-known propensity for cocrystallization (Crystal Growth & Design 2006, 6, (1), 109; Crystal Growth & Design, 2011, 11, (3), 865; Crystal Growth & Design, 2020, 20, (8), 5596, and so on) the authors should conduct controlled experiments to clarify whether the crystallization process is truly driven by the target molecules. These experiments could include using more than 10 different guest molecules instead of C60 to the target molecules to observe the influence of the molecular skeleton on the crystallization process. Additionally, the mechanism of crystallization, particularly the role of π-π interactions, is not thoroughly discussed. Molecular dynamics simulations or other theoretical calculations could help elucidate the intermolecular interactions and energy changes driving the crystallization process.

3. Universality and Generalizability of the Method:

While the authors claim the universal applicability of the 'π-trap' method for any amorphous compounds exhibiting π-π interactions, they provide insufficient experimental evidence to substantiate this claim. The manuscript focuses primarily on a single cluster, [(PhSn)4E6], and lacks experiments on other types of molecules, particularly those with significantly different structures or properties. To support the proposed universality, the authors should include experiments with a broader range of molecules, especially those that do not crystallize using conventional methods. Additionally, the manuscript does not sufficiently explore whether the method retains the crystallization-promoting capability of traditional crystal sponge methods for molecules typically used in such systems. The authors need to expand their experimental scope to demonstrate the broad applicability of their approach.

4. Selection of C60 and its Derivatives:

The choice of C60 as the co-crystallizing agent is not sufficiently explained. The authors mention C60's role in promoting crystallization but do not discuss why they selected C60 specifically over its derivatives, which could offer more diverse interaction sites. The authors should address this limitation and consider including comparative experiments with different C60 derivatives, or provide theoretical calculations to justify their choice.

Reviewer #3

(Remarks to the Author)

Through the 'π-trap' approach, the authors successfully synthesized three co-crystals by an 'unrestricted crystal sponge' approach. Compared to the typical 'crystalline sponge' method, it well overcame some limitations generally facing MOF materials, such as the size and stability. From the view of synthesis and structure, this work is important and interesting. However, the performance exploration of this study is simple, and lacks the innovation to some extent, so it is not recommended to be published in Nature Communication, a leading journal of chemistry and biology. Instead, some other journals, such as Chemical Science, Chemistry of Materials or Chemical Communication, may be suitable.

Some detailed comments are listed below:

- 1) For the synthesis, what are the yields of title compounds? Is this method universally applicable? For example, the other ligands instead of Ph and the other adamantane-like clusters, like oxide and halide. More related experiments, discussion, and results should be presented.
- 2) The characterization of title compounds, such as PXRD, TGA, FT-IR, NMR, SEM, surface potential, and flat-band potential, should be provided. Only the single crystal data may be not accepted.
- 3) Compared the C60 and the [(PhSn)4S6] precursor, what is the optical uniqueness reflected in title compounds? After all, the change in optical bandgap is a normal phenomenon.
- 4) Before dealing with UV-Vis absorption spectra, it is necessary to first clarify whether the compound is a semiconductor with a direct band gap or an indirect band gap?
- 5) The research about the performance appears to be very limited. For the combination of C60 and compound [(PhSn)4S6], what are the luminescence and nonlinear optics for the obtained materials? What is the uniqueness?
- 6) The quality of the DFT band structure is poor. The bands are flat and quite meaningless due to the quality of the calculation. The more precise method should be adopted. This part needs to be improved.
- 7) For the DFT calculations, more related discussion may be added. What conclusions can be drawn from this? What are the main sources of optical properties? What are the inspirations for subsequent work?
- 8) There are also some grammar, spelling and format mistakes in the manuscript and ESI, a double check may be needed.

Version 1:

Reviewer comments:

Reviewer #1

(Remarks to the Author)

The authors have fully addressed my previous concerns through careful revisions. Their responses and revisions have significantly improved the quality of the manuscript. The main improvements include:

1. Clear and detailed comparisons of previously calculated structures with newly determined experimental data on cluster compounds are presented, along with additional characterizations (Raman, White-light measurement).
2. Appropriate corrections have been made to terminology (e.g., MOF labeling), graphical representations (e.g., phenyl

descriptions), and optical data presentation.

3. New data on co-crystallization with other fullerenes (e.g., C₇₀, Lu₃N@C₈₀) have been included, greatly expanding the impact and relevance of the study.

I am now satisfied with the revised manuscript and recommend its publication as it is.

Reviewer #2

(Remarks to the Author)

The authors have addressed the reviewers' comments very well, resulting in a significant improvement in the quality of the manuscript. Several previously ambiguous concepts have been clarified, rendering the content more rigorous and accessible to a broad readership. In my assessment, this article now meets the publication standards of *Nature Communications*, and I recommend its acceptance.

Reviewer #3

(Remarks to the Author)

The revised manuscript now may be acceptable.

RESPONSES TO REVIEWER COMMENTS

Reviewer #1 (Remarks to the Author):

This manuscript by Dehnen and co-authors introduces a novel approach to determining the precise crystal structures of inherently amorphous cluster compounds. Their method overcomes the size limitations of traditional techniques by using π - π interactions between organic ligands on the cluster molecules and the faces of C_{60} molecules, effectively forcing the clusters into an ordered co-crystal lattice. Using this strategy, the authors not only resolve the precise crystal structures of amorphous cluster compounds of the type $[(PhSn)_4E_6]$ ($E = S, Se$) but also elucidate why these clusters do not crystallize on their own. The commercial availability of C_{60} further facilitates the adoption of this method. This discovery provides valuable insights that may contribute to future advancements in the design and development of amorphous compounds. I recommend its publication in Nature Communications following minor revisions.

Response to the comment: Thank you for your very positive assessment of our work and the helpful comments.

1. The authors' group has studied numerous adamantane-like clusters. I'm curious—were the structures of both adamantane-based clusters in this manuscript previously calculated? Additionally, have the authors compared your crystal structures with earlier theoretical models?

Response to the comment: The molecular structures of the two clusters had indeed been previously calculated – as this was the only way of getting structural information before we were able to cocrystallize them in this work. We referred to that in our introductory section on page 2: '*The structural conformations of the corresponding $[(R_3Sn)_4E_6]$ cluster molecules have been suggested by theoretical studies.*'

While some structural data had been already provided in **Supplementary Table 5** (originally **Supplementary Table 4**) this Reviewer's comment prompted us to add another table and figure (new **Supplementary Table 6** and new **Supplementary Figure 23**) to the revised version of the Supplementary Information, in which we compare all calculated (including previous calculations) and all experimental available data of the cluster molecules.

2. Have the authors investigated the generation of white light in these crystals?

Response to the comment: As the as-prepared compounds are crystalline, we did not expect the cocrystals to be white-light emitters, for which an inherently amorphous state is required. Yet, we followed the recommendation of the Reviewer and studied the nonlinear optical response.

In previous studies we found that *crystalline* samples would rather exhibit second-harmonic generation (SHG), so we studied the nonlinear response with a special focus on SHG first. Owing to possible reabsorption of light by the (red) crystals, we needed to apply an excitation wavelength above 1400 nm; however, also for the 1550 nm laser used, no SHG was observed even for high excitation densities of ~ 4 kW/cm². We therefore investigated the nonlinear response of a *pulverized* sample next, which indeed showed white-light emission like for the pure cluster molecules (and this further corroborated our assumption that the *amorphous* habitus is a prerequisite for white-light generation).

A paragraph on those results was added to the manuscript.

3. Figure 2 looks great, but the designation of the green cubic structure as "MOF" is inaccurate. In fact, it represents only the secondary building units (SBUs) rather than the full metal-organic framework. The authors should revise the labeling for precision.

Response to the comment: Thank you for pointing towards this. Actually, the inscription was meant as a title to the left-hand side of the figure, and not only as a legend of the single box – but we understand that this can be misleading. We revised the legends in the figure and also modified and expanded the caption for clarity.

4. In Figure 3, the authors use simplified grey spheres to represent phenyl groups. However, this depiction makes them resemble methyl groups, which may lead to misinterpretation. Additionally, the π - π interactions are not explicitly illustrated beyond this representation. It is suggested to retain the explicit phenyl group structure to ensure clarity and accuracy.

Response to the comment: We agree. All figures were modified accordingly.

In Figure 5, the onset of absorption for compounds 1, 2, and 3 is not clearly distinguishable. Although the authors have marked certain lines, the visibility could be improved. It is recommended to use higher-weight lines or insert a zoomed-in inset to enhance clarity.

Response to the comment: Thank you for the suggestion. We used higher-weight lines for the figure as recommended.

6. The proposed approach would be more compelling if the authors applied it to other types of clusters. Furthermore, it would be worthwhile to explore whether other fullerenes (like C₇₀, Sc₃N@C₈₀) can be incorporated using the same methodology.

Response to the comment: We are delighted to be able to add results of successful crystallization attempts with C₇₀ and Lu₃N@h-C₈₀. This further demonstrates the broader validity and applicability of the method, enhancing the relevance of the investigation. See the additions to the text and the new **Figure 7**.

7. Typos in the manuscript:

1) Page 3: "for the WLG phenomenon, and is has also not been clarified until today"

Response to the comment: Done.

2) Page 5, the caption of Figure 3: "for a single-crystalline cocrystals of C₆₀ and [(PhSn)₄S₆]"

Response to the comment: Done.

Reviewer #2 (Remarks to the Author):

In this manuscript, the authors propose the ' π -trap' method as an unrestricted crystal sponge for crystallizing inherently amorphous cluster compounds. However, I have several concerns that need to be addressed before this manuscript can be considered suitable for publication in Nature Communications.

Response to the comment: Thank you very much for your helpful comments.

1. Misapplication of the 'Crystal Sponge' Concept & Conceptual Clarity:

The manuscript refers to their method as an "unrestricted crystal sponge," which is a misapplication of the well-established concept of "crystal sponge" based on metal-organic frameworks (MOFs), as pioneered by Fujita et al. In contrast to MOF-based systems, the ' π -trap' method involving C₆₀ is fundamentally different in both structure and mechanism. This could confuse readers and should be clarified. The authors must revise their nomenclature to more accurately describe the methodology and distinguish it from MOF-based crystal sponge approaches. Additionally, the manuscript would benefit from a more detailed comparison between the ' π -trap' and conventional co-crystallization methods that use C₆₀, as this distinction is not clearly delineated. The manuscript should also explore the differences in crystallization mechanisms to justify how the π - π interactions in the proposed method are distinct from typical co-crystallization strategies.

Response to the comment: We apologize for an obviously misleading wording in our report. We did not intend to interpret our cocrystallization technique as a specific type of a 'crystal sponge' method, and therefore agree that we should not refer to it as 'unrestricted crystal sponge'. We rather intended to emphasize that our approach has no special boundary restrictions like the size of the cavities of a MOF.

We therefore eliminated this obviously confusing wording from the manuscript to sort out any misinterpretation. This included the modification of the title to read as follows in the revised version: '*The ' π -Trap' Approach for Obtaining Crystal Structure Data of Inherently Amorphous Cluster Compounds*'.

To avoid any further misunderstanding: we do not distinguish the method from other cocrystallization methods in general, but we emphasize that we need the π - π interactions to be able to cocrystallize inherently amorphous clusters that provide the opportunity to undergo such secondary interactions.

As requested, in addition to explaining why the clusters we address cannot be structurally fixed by the 'crystal-sponge' method (see the second paragraph on page 3), we elaborate in more detail on the differences of the cocrystallization using C₆₀ and other fullerenes with different types of clusters in the introductory part of the manuscript (see the additions to the third paragraphs on page 3).

2. Lack of Evidence Supporting the Mechanism of Crystallization:

The authors claim that the assembly process is induced by the target molecular skeleton, but they provide no direct evidence for this assertion. Given that C₆₀ has a well-known propensity for cocrystallization (Crystal Growth & Design 2006, 6, (1), 109; Crystal Growth & Design, 2011, 11, (3), 865; Crystal Growth & Design, 2020, 20, (8), 5596, and so on) the authors should conduct controlled experiments to clarify whether the crystallization process is truly driven by the target molecules. These experiments could include using more than 10 different guest molecules instead of C₆₀ to the target molecules to observe the influence of the molecular skeleton on the crystallization process. Additionally, the mechanism of crystallization, particularly the role of π - π interactions, is not thoroughly discussed. Molecular dynamics simulations or other theoretical calculations could help elucidate the intermolecular interactions and energy changes driving the crystallization process.

Response to the comment: Thank you for this comment and for referring to the further examples for cocrystals with C₆₀. The comment indicates that, again, we have not been clear enough in our explanations, which we tried to improve in the revised version.

Actually, our focus is rather inverse: we do not intend to claim that our cluster molecules are required for a co-crystal with C₆₀ – but the other way round. Uncountable attempts to crystallize the target molecules in other ways have remained unsuccessful since many years – with fullerene being the first molecule that enables co-crystallization at all! We therefore refer to previous cocrystals with C₆₀ (and we were happy to add the references given in this comment to the list). We emphasize that C₆₀ and the π - π interactions between the two components of the cocrystal are urgently necessary required to allow the clusters to order – see also our answer to you comment no. 1. **Figure 4**, and new **Figure 7** for newly-added compounds (see also our answers to your comments 3. and 4.), were created to give structural details including the measures for the π - π interactions.

Following the suggestion of the Reviewer, we have performed additional calculations to model the electrostatic potential of the adamantane-type clusters as isolated molecules and in the cocrystals. The calculations suggest that the core···core interactions are at least as important as the substituent···substituent interactions, as expected for molecules which do not form ordered crystals. The latter can only be formed by the presence of the C₆₀ molecules. These force the cluster molecules into secondary interactions by defining high-potential regions in the crystal. Thus, while the C₆₀ molecules do not substantially alter the potential differences within the adamantane-type cluster molecules themselves, they provide a template for the crystallization.

We hope that the additions to the manuscript were helpful in clarifying this. If not, please let us know, so that we can improve this aspect even further.

3. Universality and Generalizability of the Method:

While the authors claim the universal applicability of the ‘ π -trap’ method for any amorphous compounds exhibiting π - π interactions, they provide insufficient experimental evidence to substantiate this claim. The manuscript focuses primarily on a single cluster, [(PhSn)₄E6], and lacks experiments on other types of molecules, particularly those with significantly different structures or properties. To support the proposed universality, the authors should include experiments with a broader range of molecules, especially those that do not crystallize using conventional methods. Additionally, the manuscript does not sufficiently explore whether the method retains the crystallization-promoting capability of traditional crystal sponge methods for molecules typically used in such systems. The authors need to expand their experimental scope to demonstrate the broad applicability of their approach.

Response to the comment: Although we intended to write an article about a proof-of-principle study, we agree that further examples are helpful to proof the broader applicability of the approach. As requested, we included additional examples for the method in the revised manuscript.

Please note though that we do not claim, that this method is generally applicable to all kinds of compounds. As explained in our work, the method is tailor-made for (1) *inherently non-crystalline* (amorphous or liquid) compounds that (2) are *capable of undergoing π - π interactions*. Many other – crystalline or non-crystalline – compounds were already reported to cocrystallize with fullerene, to which we also refer in our manuscript. We could also cocrystallize further clusters that we can also obtain without the help of C₆₀ though. So, the special aspect of this study is the crystallization of cluster compounds that are inherently non-crystalline. To the best of our knowledge, most of these clusters are of the adamantane type. So, to fit the scope of our work, we expanded the study to another cluster of this type, [(NpSn)₄S₆], which can also be cocrystallized with larger fullerenes, C₇₀ and Lu₃N@C₈₀ (see also our answer to your comment 4.), thus allowing for a selective crystallization of this cluster from a mixture. We have created a new paragraph on this addition and a new figure (**Figure 7**).

4. Selection of C60 and its Derivatives:

The choice of C₆₀ as the co-crystallizing agent is not sufficiently explained. The authors mention C₆₀'s role in promoting crystallization but do not discuss why they selected C₆₀ specifically over its derivatives, which could offer more diverse interaction sites. The authors should address this limitation and consider including comparative experiments with different C₆₀ derivatives, or provide theoretical calculations to justify their choice.

Response to the comment: We agree – and we have therefore added studies with larger fullerene C₇₀ and Lu₃N@C₈₀ that undergo π - π interactions preferably with clusters exhibiting larger aromatic systems, such as naphthyl groups in [(NpSn)₄S₆], see the additions to the text and new **Figure 7**. We are happy to add that the application of the larger fullerenes additionally serves to selectively crystallize [(NpSn)₄S₆], as clusters exhibiting a less-extended π system, such as provided by phenyl substituents do only cocrystallize with C₆₀.

Theoretical work was already included in the originally submission, please see **Figure 6** and the accompanying text. It was shown that the interactions are not largely affecting the clusters electronic structures, such as typical for secondary interactions. For even more insight into the nature of the cocrystals, we recorded Raman spectra, which were added to revised **Figure 5** and discussed in that context.

Reviewer #3 (Remarks to the Author):

Through the ‘ π -trap’ approach, the authors successfully synthesized three co-crystals by an ‘unrestricted crystal sponge’ approach. Compared to the typical ‘crystalline sponge’ method, it well overcame some limitations generally facing MOF materials, such as the size and stability. From the view of synthesis and structure, this work is important and interesting. However, the performance exploration of this study is simple, and lacks the innovation to some extent, so it is not recommended to be published in Nature Communication, a leading journal of chemistry and biology. Instead, some other journals, such as Chemical Science, Chemistry of Materials or Chemical Communication, may be suitable.

Response to the comment: We thank the Reviewer for their helpful comments.

Some detailed comments are listed below:

1) For the synthesis, what are the yields of title compounds? Is this method universally applicable? For example, the other ligands instead of Ph and the other adamantane-like clusters, like oxide and halide. More related experiments, discussion, and results should be presented.

Response to the comment: We apologize for not having indicated the yields in the original documents. They were added to the revised manuscript. The applicability is tailor-made for compounds fulfilling the following characteristics: they (1) need to be *inherently non-crystalline* (amorphous or liquid) and (2) they need to be *capable of undergoing π - π interactions*. We have added some further cluster examples to corroborate this. We emphasize that this method is the only one proven to work for cluster compounds with characteristics (1) and (2). For this, we refrained from adding examples of smaller, i.e. non-cluster, molecules, as for these compounds, other methods apply as well.

2) The characterization of title compounds, such as PXRD, TGA, FT-IR, NMR, SEM, surface potential, and flat-band potential, should be provided. Only the single crystal data may be not accepted.

Response to the comment: We understand that the provision of as many analytical proofs as possible for new compounds is desirable. At the same time, however, we refer to the fact that other works have been published in Nature Communications and other highest-ranking journals without providing the full set asked for by this Reviewer. In our original submission, we have reported single-crystal structures, UV-Vis spectra, first-principles calculations of the atomic and electronic structure as well as of the optical response. In our revision, we added SEM images, a study of the nonlinear response and Raman spectra. We additionally calculated electrostatic potentials for clusters $[(\text{PhSn})_4\text{S}_6]$ and $[(\text{PhSn})_4\text{Se}_6]$ – both as isolated molecules and within compounds **1** and **3** (see new **Supplementary Figure 24** and accompanying text).

A PXRD diagram was recorded for single crystals of compound **1**, which took us several months to collect owing to the limited yield. We are happy to provide it here (**Figure RL1**) to demonstrate a good agreement with the simulation – in spite of a limited signal-to-noise-ratio that is a consequence of the small amount of crystals. As we are exclusively talking about the results of single-crystal structure elucidations in our work, we assume that PXRD is not mandatory in this context for all six compounds.

We hope for this Reviewer’s understanding that we have decided that further analysis would not provide any additional insights that are critical for the purpose and scope of this study. Moreover, as we do not have access to the latter two methods, addition of the measurements would have led to a significant delay in re-submission, which was not desirable.

Figure RL 1: Powder-X-ray diffraction (PXRD) diagram of compound **1** along with its simulation.

3) Compared the C60 and the $[(\text{PhSn})_4\text{S}_6]$ precursor, what is the optical uniqueness reflected in title compounds? After all, the change in optical bandgap is a normal phenomenon.

Response to the comment: Please note that the main focus of our work is not to produce new compounds with optical peculiarities, but to elucidate the structural properties of compounds which inherently feature a peculiar optical response. This issue addresses clusters like $[(\text{PhSn})_4\text{S}_6]$, the structural data of which have been inaccessible, but are urgently needed to gain insight into the reasons for the phenomena. As we outline in the newly expanded introductory section, numerous theoretical and simulation-based predictions had been made in the past as to why the clusters are inherently amorphous, and as to why this habitus is a necessary precondition for their extreme nonlinear optical response.

As the most important output of our study, the long sought-for access to crystal structure data, we can finally confirm these theoretical predictions, which, thanks to our work, are no longer speculative, and provide even more insight! This is an enormous step forward in understanding the uniqueness of amorphous molecular materials with extreme non-linear optical properties, as the result of a collaborative effort that has been funded by the German Research Foundation in the framework of Research Unit FOR 2824 for the past eight years and has sparked many further research activities.

4) Before dealing with UV-Vis absorption spectra, it is necessary to first clarify whether the compound is a semiconductor with a direct band gap or an indirect band gap?

Response to the comment: We agree that this information was missing in the previous version of the manuscript. We added Tauc plots to the revised Supporting Information as new **Supplementary Figure 21**, which suggest that the compounds exhibit a direct band gap; please note though that, due to the flat band dispersion, the difference between direct and indirect bandgap is below 0.1 eV.

5) The research about the performance appears to be very limited. For the combination of C₆₀ and compound [(PhSn)₄S₆], what are the luminescence and nonlinear optics for the obtained materials? What is the uniqueness?

Response to the comment: We agree that more information on the performance of the cocrystals would be desirable. We have therefore added and discussed non-linear response spectra and Raman spectra (see revised **Figure 5**). Again, please note that the uniqueness is laid in the methodology to get access to structural data of unique amorphous compounds – thus, a contradiction in itself. The famous ‘crystal sponge’ method introduced by Fujita in his seminal work in Nat. Chem. 2010, Nature 2013, and Nat. Protoc. 2014 also addressed this aspect, while the chemical or physical properties were not in the focus of the publication. Nevertheless, we are happy to report insight into the electronic structure based on first-principles calculations, indicating that in this case, blending of the two compounds does not affect their individual molecular electronic structures, but leads to the insertion of localized electronic states from the fullerene molecules in the center of the energy gap of the cluster molecules’ electronic structure.

6) The quality of the DFT band structure is poor. The bands are flat and quite meaningless due to the quality of the calculation. The more precise method should be adopted. This part needs to be improved.

Response to the comment: We thank the Reviewer for his/her suggestion. At his point it is important to distinguish between the accuracy of the method and the inherent nature of the electronic band structure of the investigated compounds.

Concerning the accuracy of the method, we remark that the unit cells of the investigated crystals contain 560, 852, and 1084 atoms, respectively. The calculation of the electronic band structures for systems of this size within Density Functional Theory + vdW correction schemes is already rather challenging, and can only be accomplished with a high computational effort on supercomputers. Electronic structure calculations beyond DFT (e.g., including many-body effects in a pure quantum chemical manner) for systems containing 1000+ atoms are unfortunately beyond the actual computational capabilities.

Nonetheless, in order to estimate the effect of more refined computational approaches on the investigated systems, we have performed atomic and electronic structure calculations within hybrid-DFT in the HSE06 formulation for the parent molecules [(PhSn)₄S₆] (**A**), [(PhSn)₄Se₆] (**B**) and C₆₀. The atomic structures of the compounds do not significantly deviate from the DFT calculated geometry. The averaged bond distances calculated by hybrid-DFT are about 1% smaller than the corresponding distances calculated by DFT. The calculated values (all lengths in Å) are listed in **Table RL1**:

Table RL 1: Averaged bond lengths (in Å) in [(PhSn)₄S₆], [(PhSn)₄Se₆] and C₆₀ as isolated molecules as calculated by DFT with semi-local XC functionals (PBE) or by hybrid-DFT (HSE06).

	A [(PhSn)₄S₆]		B [(PhSn)₄Se₆]		C₆₀
	Sn-S	C-C	Sn-Se	C-C	C-C
PBE	2.429	1.398	2.560	1.398	1.435
HSE06	2.399	1.390	2.524	1.390	1.426
Diff.	-1.2%	-0.6%	-1.4%	-0.6%	-0.6%

The largest absolute structural deviation is calculated for the Sn–Se bond length in the [(PhSn)₄Se₆] molecule and amounts to 1.43%. All other deviations are smaller. The HSE06 calculated coordinates have been included in the SI as well.

Concerning the electronic states, including Hartree-Fock exchange in the short-range part of the exchange term opens up the HOMO-LUMO gap by 1.4 eV and 1.3 eV in **A** and **B**, and by 0.7 eV in C_{60} , respectively (see **Figure RL2**). Consequently, the C_{60} related electronic states still fall well within the HOMO-LUMO gaps of $[(PhSn)_4S_6]$ and $[(PhSn)_4Se_6]$, exactly as predicted within DFT (actually, even better), and none of the conclusions in the manuscript is affected.

Furthermore, we would like to note that the electronic band structures of the investigated compounds are not flat because of a poor quality of the computational approach, but because the parent molecules are electronically rather decoupled within the crystals, even considering dispersion forces!

With a few known exceptions, where refined many-body dispersion approaches are required to capture the fine intramolecular interactions between the molecules in the crystal (see L. Kroonik and A. Tkachenko, *Acc. Chem. Res.* 2014, 47, 11, 3208–3216 and Gregory J. O. Beran, *Chem. Rev.* 2016, 116, 9, 5567–5613), a flat band dispersion is a common feature in molecular crystals. The electronic states of the crystals **1**, **2**, and **3** strongly resemble those of the parent molecules and this does *not* depend on the quality of the computational approach.

The flat band structure mentioned by the Reviewer was also related to the rather large energy interval chosen for the representation. As a further step toward a more insightful representation of the band structures, we have optimized the band structure plots by highlighting the states with a strong C_{60} character and focusing on a more restricted region about the HOMO-LUMO gap of the parent molecular clusters **A** and **B**. The new plots allow to appreciate the band dispersion to a larger extent.

Figure RL2: Electronic bandgap (in eV) of $[(PhSn)_4S_6]$, $[(PhSn)_4Se_6]$ and C_{60} as isolated molecules calculated by DFT-PBE (left) and hybrid-DFT (HSE06, right).

7) For the DFT calculations, more related discussion may be added. What conclusions can be drawn from this? What are the main sources of optical properties? What are the inspirations for subsequent work?

Response to the comment: According to the suggestion of the Reviewer, we have expanded the documentation and discussion of the DFT results. We have added new aspects to the discussion in the main document and we have added another table, as well as new figures and comments to the revised Supporting Information (new **Supplementary Table 6**, new **Supplementary Figure 23**, and new **Supplementary Figure 24**).

In new **Supplementary Table 6** and new **Supplementary Figure 23**, we compare the structures of clusters $[(PhSn)_4S_6]$ and $[(PhSn)_4Se_6]$ in the gas phase, as free-standing pairs, and in the (calculated and experimentally observed) crystals. The rather small dependence of the bond lengths and bond angles on the environment strongly suggests that the C_{60} molecules merely provide a template for the crystallization of the compounds, without profoundly affecting their structural parameters.

The minor impact of the environment on the molecular cluster structures is in further agreement with the vanishing coupling of the electronic states of C_{60} and of the adamantane-based clusters. In the framework of the previous answers, we have calculated the atomic and electronic structure of the single crystal components to show that applying more refined computational methods including exact-exchange does not modify any of the conclusions concerning the electronic and atomic structure (see also previous answer). We added this information to the revised manuscript.

We also calculated the electrostatic molecular potential for the $[(PhSn)_4S_6]$ and $[(PhSn)_4Se_6]$ clusters, both for the free-standing molecules and as extracted from the calculated crystal structures of **1** (as an example of **1** and **2**) and **3**, see new **Supplementary Figure 24**. This allowed to gather information about their tendency to aggregate in a rather unordered way in the absence of C_{60} , but in an ordered way in its presence. The results corroborate that the electrostatic (rather isotropic) core...core interaction between clusters of both $[(PhSn)_4S_6]$ or $[(PhSn)_4Se_6]$ is rather strong in the absence of C_{60} , which confirms their tendency to aggregate rather arbitrarily instead of crystallizing in ordered structures. Yet, the template provided by the C_{60} molecules allows for the formation of an ordered, periodic structure by means of π - π interactions between the C_{60} surface and the aromatic organic substituents of the clusters. This information was also added to the revised manuscript.

The main sources of the optical properties are not investigated in this work, which deals with the PI-Trap method and the synthesis of periodic molecular crystals from molecular clusters which otherwise resist any attempt to crystallize. Yet, in previous investigations we could demonstrate that the optical nonlinearities in the wide family of tetraphenyl hetero-adamantanes have their origin in electronic transitions occurring within the delocalized orbitals of the substituents. Compounds **A** and **B** are no exception.

Finally, we would like to provide a comment concerning the value of the present investigation in inspiring new studies, which we probably did not remark enough in the original version of the manuscript. While cocrystallization with C_{60} is not exclusive of the present work, the employment of fullerenes to obtain molecular clusters from polyhedral molecules which are intrinsically amorphous and have resisted any previous attempt to crystallize is the new aspect of the present report. We propose a method that makes use of the cocrystallization of cluster molecules with cheap and commercially available fullerenes. On the one side, the described procedure allows to gather experimental data on the atomic structure of molecular clusters that are otherwise not experimentally accessible. This is a crucial advance, as the atomic structure is strongly related to the physical and chemical properties of the investigated compounds. In the particular case of clusters **A** and **B**, e.g., the atomic structure influences the nature of the rather uncommon nonlinear optical response, which is still under intense investigation and thus requires such structural insights. On the other hand, the proposed method opens the door for a multitude of further studies and developments. Many variations of the procedure can be conceived, including cocrystallization with other fullerenes beyond C_{60} , e.g., C_{70} and $Lu_3N@C_{80}$, as showcased in the revised version of our manuscript. This is a new pathway in chemical synthesis, which is inspired by the present work and expects to be explored.

8) There are also some grammar, spelling and format mistakes in the manuscript and ESI, a double check may be needed.

Response to the comment: We thank this Reviewer for pointing to the errors that were corrected during revision. These (minor) corrections are not highlighted by yellow background.

We would like to thank the Editor and the Reviewers for their careful inspection of our work and the helpful suggestions!